# Subcellular analysis of pigeon hair cells implicates vesicular trafficking in cuticulosome formation and maintenance

Simon Nimpf[1], Erich Pascal Malkemper[1], Mattias Lauwers[1], Lyubov Ushakova[1], Gregory Nordmann[1], Andrea Wenninger-Weinzierl[1], Thomas R Burkard[1], Sonja Jacob[2], Thomas Heuser[2], Guenter P Resch[3], David A Keays[1]*

[1]Research Institute of Molecular Pathology, Vienna Biocenter, Vienna, Austria; [2]Electron Microscopy Facility, Vienna BioCenter Core Facilities GmbH, Vienna, Austria; [3]Nexperion e.U., Vienna, Austria

**Abstract** Hair cells are specialized sensors located in the inner ear that enable the transduction of sound, motion, and gravity into neuronal impulses. In birds some hair cells contain an iron-rich organelle, the cuticulosome, that has been implicated in the magnetic sense. Here, we exploit histological, transcriptomic, and tomographic methods to investigate the development of cuticulosomes, as well as the molecular and subcellular architecture of cuticulosome positive hair cells. We show that this organelle forms rapidly after hatching in a process that involves vesicle fusion and nucleation of ferritin nanoparticles. We further report that transcripts involved in endocytosis, extracellular exosomes, and metal ion binding are differentially expressed in cuticulosome positive hair cells. These data suggest that the cuticulosome and the associated molecular machinery regulate the concentration of iron within the labyrinth of the inner ear, which might indirectly tune a magnetic sensor that relies on electromagnetic induction.

DOI: https://doi.org/10.7554/eLife.29959.001

**\*For correspondence:**
keays@imp.ac.at

**Competing interests:** The authors declare that no competing interests exist.

## Introduction

Hair cells are located in defined sensory epithelia within the vertebrate inner ear, enabling the transduction of sound, motion, and gravity into neuronal impulses (*Torres and Giráldez, 1998*). This is mediated by mechanical deflection of stereocilia, which are bristle-like projections that emerge from the apical surface of the cell. Displacement of stereocilia opens cation channels via filamentous structures called tip-links, resulting in cation influx, and subsequently neurotransmitter release (*Pickles et al., 1984*). Stereocilia are composed of tightly packed parallel filaments of f-actin, which taper at their base forming a rootlet that inserts into the cuticular plate (*Pollock and McDermott, 2015*). The latter is a meshwork of actin at the base of the stereocilia that has a characteristic U-shape and is believed to provide stability to the hair bundle. The kinocillium, with its distinctive core of microtubules, also inserts into the cuticular plate serving as an important guidepost for stereocilia development and orientation (*Montcouquiol et al., 2003*). These unique structures were initially identified using transmission electron microscopy, but have been further characterized using electron tomographic methods. Insight into the molecules that are required to construct them has resulted primarily from genetic and transcriptomic studies in various model systems (*Mburu et al., 2003*; *Kitajiri et al., 2010*; *Antonellis et al., 2014*; *Liu et al., 2014*).

Following the studies of Dickman and colleagues, who reported that neurons within the vestibular nuclei of pigeons are responsive to weak magnetic stimuli, we conducted a histological analysis of the inner ear of birds employing the Prussian blue stain which labels ferric iron (*Wu and Dickman, 2011, 2012*). This led to the discovery of an iron-rich organelle in the cuticular plate of avian hair

cells, which was termed the 'cuticulosome' (*Lauwers et al., 2013*). Electron microscopy revealed that cuticulosomes are approximately 400–500 nm in diameter and 25% are encapsulated by a visible membrane. Each cuticulosome consists of tens of thousands of ferritin-like granules, which have a distinct paracrystalline organization in some but not all cases. Cuticulosomes can be found in all sensory epithelia of the inner ear, however, they are most prevalent in the avian equivalent of the organ of Corti. In this structure, known as the basilar papilla, approximately 30% of hair cells are cuticulosome positive. A phylogenic analysis of adult birds revealed that cuticulosomes are an evolutionary conserved feature in a broad range of avian species (e.g. zebra finches, ducks, chickens and ostriches) but are absent in mice, humans and fish. The function of this curious organelle is currently unknown. While it may play a role in the magnetic sense of aves or in the stabilization of stereocilia (*Wu and Dickman, 2011*; *Jandacka et al., 2015*), its presence might simply reflect the late onset accumulation of iron which has been described in some neurodegenerative disorders (*Xie et al., 2014*).

Accordingly, we set out to investigate the developmental maturation of cuticulosomes in pigeons, as well as the molecular and subcellular architecture of cuticulosome positive hair cells. Exploiting electron microscopy and histological analysis we show that cuticulosomes develop rapidly after hatching, in a process that involves vesicle fusion and nucleation of ferritin nanoparticles within the cuticular plate. Coupled with a comparative transcriptomic analysis of hair cells with and without cuticulosomes our results emphasize the importance of vesicular function in the formation and maintenance of this iron-rich structure.

## Results

### Cuticulosomes form rapidly after birds hatch

To investigate when cuticulosomes are formed during development, we employed the chemical stain Prussian blue (PB) which stains ferric iron ($Fe^{3+}$) bright blue in color (*Figure 1a–b*). Coronal sections of the basilar papilla and the lagenar macula from pigeons of different ages (1 day, 8 days, 16 days, 30 days and 1 year) were stained and quantitated using light-microscopy (*Figure 1—figure supplement 1*). In 1 day old birds only a small percentage of hair cells in the basilar papilla were cuticulosome positive (2.46 ± 0.95%, n = 4 birds), whereas they were plentiful at 8 days (23.66 ± 3.03%, n = 3 birds), 16 days (31.19 ± 2.86%, n = 6 birds), 30 days (36.9 ± 3.56%, n = 5 birds), and 1 year of age (24.92 ± 6.74%, n = 3 birds) (*Figure 1c* and *Figure 1—source data 1*). Consistent with our previous results there were few cuticulosome positive hair cells in the lagena, with a moderate increase from 0.06 ± 0.019% (1 day, n = 4 birds), 0.57 ± 0.22% (8 days, n = 3 birds), 0.95 ± 0.17% (16 days, n = 6 birds), 1.72 ± 0.51% (30 days, n = 5 birds) to 3.43 ± 1.77% in 1 year old pigeons (n = 3 birds) (*Figure 1d* and *Figure 1—source data 1*). We mapped the distribution of PB positive cells along the length of the cochlear duct for each time point. At day 1 and day 8 the distribution of PB positive hair cells appears stochastic, however, at subsequent time points we observed a noticeable enrichment of cuticulosomes in the distal regions of the basilar papilla (*Figure 1—figure supplements 2–6*).

Next, we asked where in the cuticular plate do these iron-organelles form by undertaking whole mount Prussian blue staining on intact basilar papillae, capturing images of the epithelium from a superior perspective (*Figure 1b*). Strikingly, almost all cuticulosomes were located opposite the stereocillia. We measured the length of the cell along its apical surface and recorded the position of the cuticulosome along that vector. This revealed that the position of the cuticulosome gradually changes with time, from a lateral localization, close to the border of the cell in 1 day old birds, to a more medial localization closer to the stereocilia, in 1 year old birds (*Figure 1e*). We then examined the depth of the cuticulosome within the cell, by analysing coronal sections of the basilar papilla. The height of the cell was measured from its apical to basal end (excluding the hair cell bundle), and the position of the cuticulosome was plotted. We observed that in 1 day old birds the cuticulosome is closest to the apical cell surface, and that it gradually adopts a deeper position within the cuticular plate over the course of a year (*Figure 1f*). Taken together these data show that cuticulosomes form rapidly after hatching, that they are not the product of late-onset iron accumulation and that their localization within the cuticular plate changes from a lateral-apical to a medio-basal position as they mature.

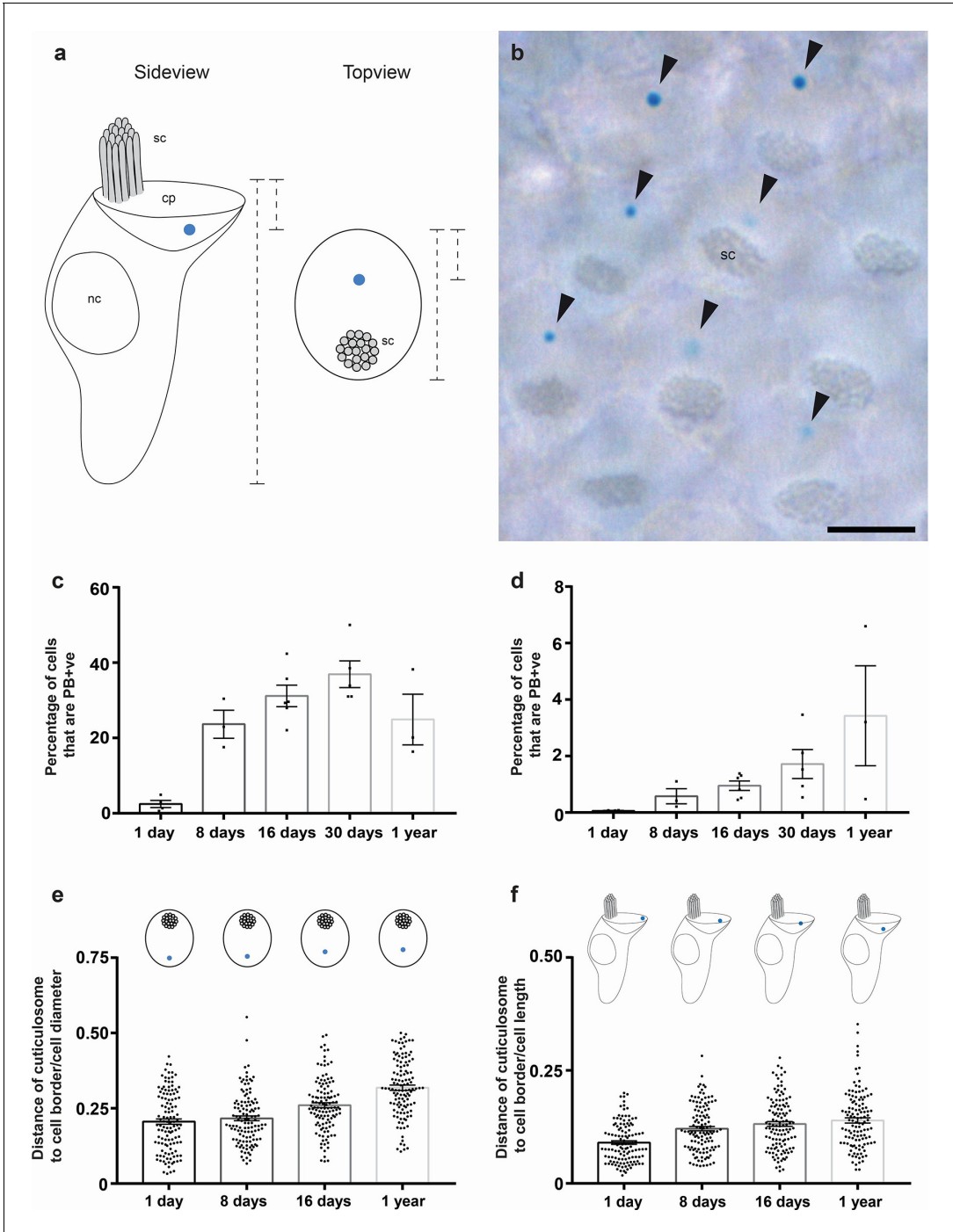

**Figure 1.** Development and cellular localization of cuticulosomes. (a) Schematic representation showing the localization of iron-rich organelles in hair cells viewed from the side and the top. The stereocilia (sc), the nucleus (nc) and the cuticular plate (cp) are labelled. Dashed lines indicate measurements that were taken for (e and f) (b) Basilar papilla whole mount from a one year old bird that was stained with Prussian blue, showing a top-down view on hair cells with stereocilia (sc) and PB +ve iron-rich organelles highlighted with arrowheads. Scale bar represents 5 µm. (c) Graph showing the percentage of hair cells that contain Prussian blue positive cuticulosomes in the basilar papilla of 1 day old pigeons (2.46%, n = 4 birds, n = 36644 cells), 8 day old pigeons (23.66%, n = 3 birds, n = 37292 cells), 16 day old pigeons (31.19%, n = 6 birds, n = 86558 cells), 30 day old pigeons (36.9%, n = 5 birds, n = 64668 cells) and 1 year old pigeons (24.9%, n = 3 birds, n = 33312 cells). Dots show the percentage of PB +ve hair cells in individual birds. (d) Graph showing the percentage of hair cells that contain Prussian blue positive cuticulosomes in the lagena of 1 day old pigeons (0.06%, n = 4 birds, n = 24888 cells), 8 day old pigeons (0.57% n = 3 birds, n = 25396 cells), 16 day old pigeons (0.95%, n = 6 birds, n = 103196 cells), 30 day old pigeons (1.72%, n = 5 birds, n = 79536 cells) and 1 year old pigeons (3.43%, n = 3 birds, n = 57264 cells). Dots show the percentage of PB +ve hair cells in individual birds. (e) Graph showing the distance of cuticulosomes from the lateral cell border as a ratio of the cell width at four developmental time

*Figure 1 continued on next page*

*Figure 1 continued*

points (n = 3–4 birds and n = 120 individual measurements per time point). Dots show ratio for individual cells. (f) Graph showing the distance of cuticulosomes from the apical cell border as a ratio of the cell height at four developmental time points (n = 3 birds per timepoint, n = 120 individual measurements per time point). Dots show ratio for individual cells. Error bars for all graphs show the mean ± SEM.

DOI: https://doi.org/10.7554/eLife.29959.002

The following source data and figure supplements are available for figure 1:

**Source data 1.** Quantitation of Prussian blue positive hair cells in the basilar papilla and lagena macula.

DOI: https://doi.org/10.7554/eLife.29959.009

**Figure supplement 1.** Developmental time line of iron-rich organelles.

DOI: https://doi.org/10.7554/eLife.29959.003

**Figure supplement 2.** Quantitation and distribution of iron-rich organelles in the cochlear duct of 1 day old pigeons.

DOI: https://doi.org/10.7554/eLife.29959.004

**Figure supplement 3.** Quantitation and distribution of iron-rich organelles in the cochlear duct of 8 day old pigeons.

DOI: https://doi.org/10.7554/eLife.29959.005

**Figure supplement 4.** Quantitation and distribution of iron-rich organelles in the cochlear duct of 16 day old pigeons.

DOI: https://doi.org/10.7554/eLife.29959.006

**Figure supplement 5.** Quantitation and distribution of iron-rich organelles in the cochlear duct of 30 day old pigeons.

DOI: https://doi.org/10.7554/eLife.29959.007

**Figure supplement 6.** Quantitation and distribution of iron-rich organelles in the cochlear duct of 1 year old pigeons.

DOI: https://doi.org/10.7554/eLife.29959.008

## The subcellular architecture of cuticulosomes during development

To gain insight into the subcellular architecture of cuticulosomes during development, we employed transmission electron microscopy (TEM) on basilar papillae isolated from pigeons aged 1 day, 8 days, 16 days, 30 days and 1 year. At 1 day of age we observed that 100% of organelles were in the process of formation (n = 3 birds, n = 4 cuticulosomes), which gradually decreased to 34.6% at 8 days (n = 3 birds, n = 26 cuticulosomes), 22.2% at 16 days (n = 3 birds, n = 45 cuticulosomes) and 17.6% at 30 days (n = 3 birds, n = 17 cuticulosomes) (*Figure 2* and *Figure 2—source data 1*). In contrast, only 7% of organelles in 1 year old pigeon hair cells appeared to be incomplete (*Lauwers et al., 2013*). During this maturation the ferritin-like granules that make up the structure were initial diffuse, but became more dense and structured. In 1 day old birds (n = 3 birds, n = 4 cuticulosomes), we did not observe any evidence of paracrystalline ordering in contrast to 16 day old birds (n = 3 birds, n = 26 cuticulosomes) where ordering of granules was observed in 17.8% of cases (*Figure 2—source data 1*). This maturation coincided with an increase in cuticulosome size from 296.4 nm ± 69.2 nm in 1 day old birds (n = 3 birds, n = 4 cuticulosomes), to 387.9 ± 29 nm in 8 day old birds (n = 3 birds, n = 26 cuticulosomes), 580.7 nm ± 21.7 nm in 16 days old birds (n = 3 birds, n = 45 cuticulosomes), and 514.3 ± 38 nm in 30 day old birds (n = 3 birds, n = 17 cuticulosomes) (*Figure 2k*). The percentage of cuticulosomes surrounded by lipid membranes was similar at 1 day, 8 days, 16 days, 30 days and 1 year (*Figure 2—source data 1*). In addition we quantitated the number of cuticulosomes that were associated with vesicles at the periphery of the structure. We found that at 1 day of age 100% of cuticulosomes were associated with vesicles (n = 3 birds, n = 4 cuticulosomes), 46.15% at 8 days (n = 3 birds, n = 26 cuticulosomes), 24.44% at 16 days (n = 3 birds, n = 45 cuticulosomes), 52.94% at 30 days (n = 3 birds, n = 17 cuticulosomes) and 25% at 1 year (*Lauwers et al., 2013*) (*Figure 2—source data 1*). In some instances we observed streams of vesicles that appeared to converge on incomplete cuticulosomes (See *Figure 2c* and inset).

To investigate this further we performed electron tomography on 8 day old birds, preparing 200–250 nm thick resin embedded sections of the basilar papilla that were analyzed on a Polara transmission electron microscope with a −60° to +60° tilt series. Segmentation of the ferritin-like nanoparticles, vesicular structures, and the cell membrane allowed the generation of 3D tomographic models of the developing cuticulosome (n = 4 cuticulosomes, n = 3 birds) (*Figure 3* and *Figure 3—videos 1–4*). This suggests that at 8 days post hatching cuticulosomes are decorated by numerous vesicles that range in size from 30 nm to 200 nm (*Figure 3a–l*). Some, but not all, of these vesicles contain ferritin-like nanoparticles and appear to be fusing with the larger organelle (*Figure 3e–f,h–i*). We again observed the trafficking of vesicles from the periphery of the cuticular plate to the cuticulosome and in rare instances invaginations of the apical membrane indicative of an endocytotic/

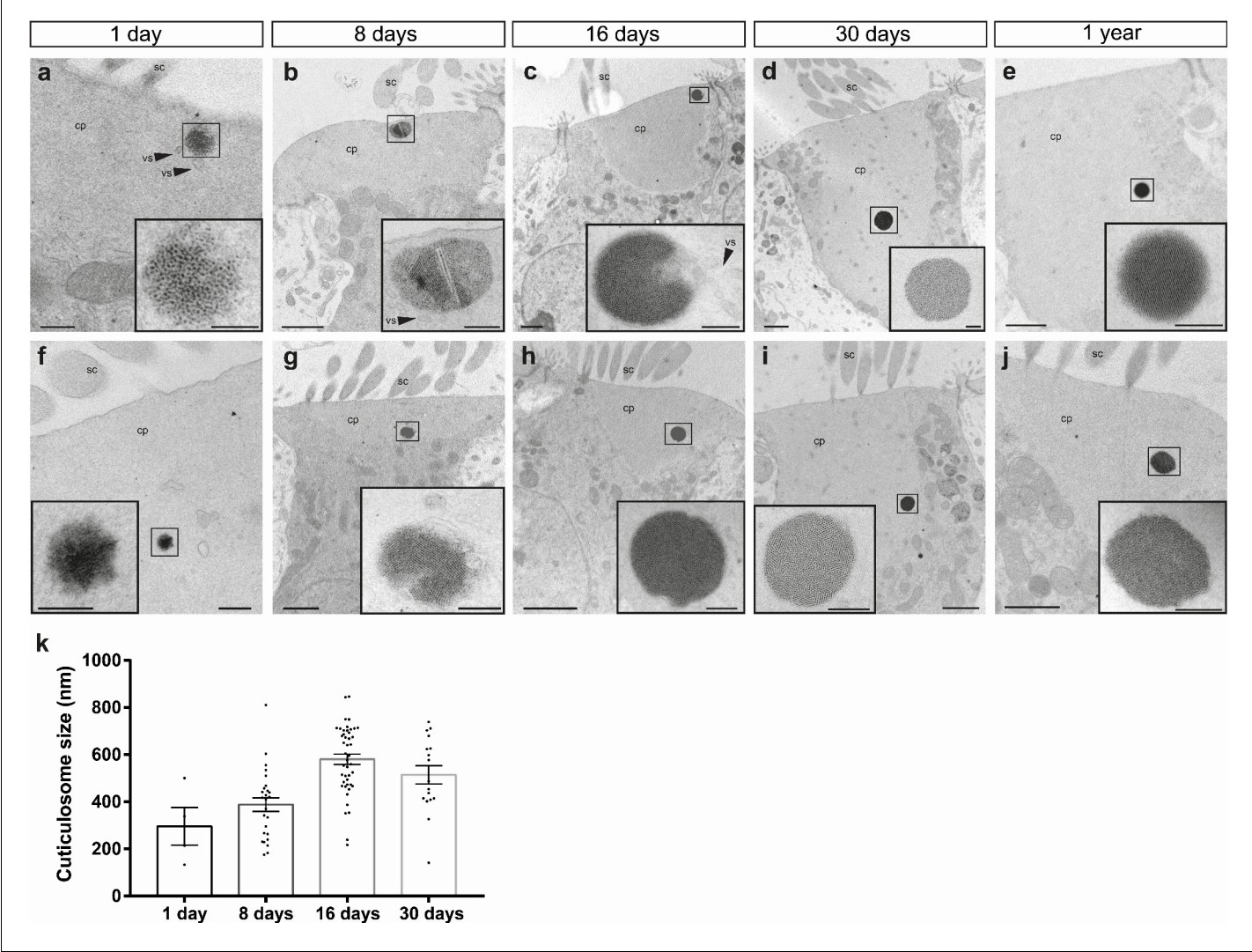

**Figure 2.** Subcellular architecture of iron-rich organelles during development. (**a–h**) Electron micrographs showing cuticulosomes in the cuticular plate (cp) apically located beneath the stereocilia (sc) in hair cells from the basilar papilla of 1 day (**a** and **f**), 8 day (**b** and **g**), 16 day (**c** and **h**), 30 day (**d** and **i**) and 1 year old pigeons (**e** and **j**). At hatching cuticulosomes are amorphous aggregations of ferritin nanoparticles, that increase in size, density and organisation as they mature. In some cases, cuticulosomes are organized in paracrystalline arrays (**e** and inset), surrounded by membranes (**c**, **h** and insets) and decorated with vesicles (vs) (**a**,**b**, **c**, and insets). (**i**) Graph showing the mean diameter of cuticulosomes from the basilar papilla of 1 day (296.4 ± 69.2 nm, n = 3 birds, n = 4 cuticulosomes), 8 day (387.9 ± 29.0 nm, n = 3 birds, n = 26 cuticulosomes), 16 day (580.7 ± 21.7 nm, n = 3 birds, n = 45 cuticulosomes) and 30 day old birds (514.3 ± 38 nm, n = 3 birds, n = 17 cuticulosomes). Dots show measurements for individual cuticulosomes. Error bars show the mean ± SEM. Scale bars represent 0.25 μm in **a** and **f**, 1 μm in **b**, **c**, **d**,**e**, **g**, **h**, **i** and **j**, 200 nm in insets of **b**, **c**, **d**, **e**, **g**, **h**, **i** and **j** and 100 nm in insets of **a** and **f**.

DOI: https://doi.org/10.7554/eLife.29959.010

The following source data and figure supplement are available for figure 2:

**Source data 1.** Properties of cuticulosomes at different ages.
DOI: https://doi.org/10.7554/eLife.29959.012

**Figure supplement 1.** Density analysis of the cuticular plate.
DOI: https://doi.org/10.7554/eLife.29959.011

exocytotic event (*Figure 3k–l*). We did not observe any actin free tunnels within the cuticular plate, which would serve as subcellular conduits to the cuticulosome. We quantified the staining intensity of electron micrographs along 200 nm lines that extend radially at 45° angles from the periphery of cuticulosomes (n = 3 birds, n = 8 cuticulosomes). This revealed, that with the exception of vesicles,

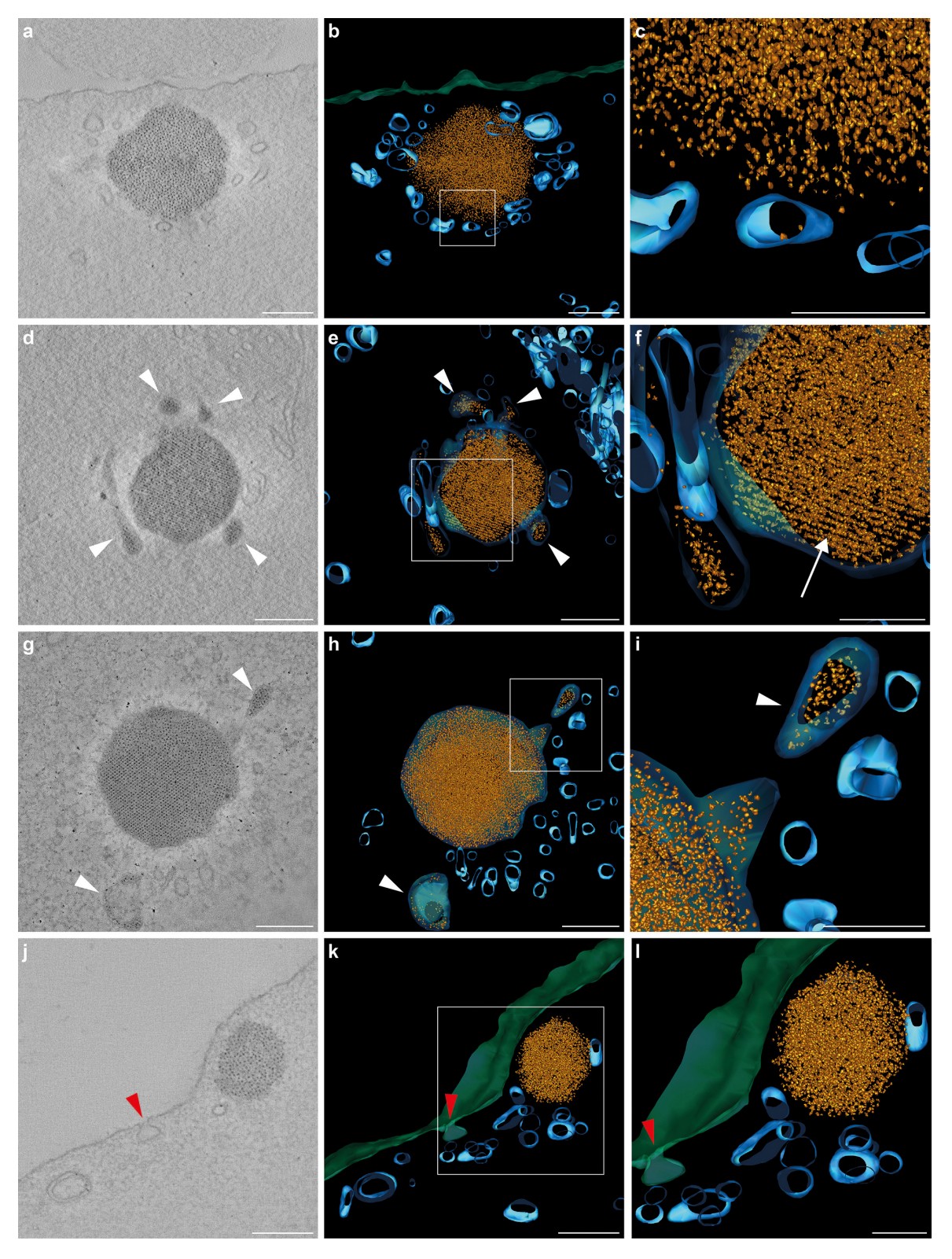

**Figure 3.** Tomograms and three-dimensional reconstructions of cuticulosomes in 8 days old pigeons. (**a, d, g, j**) Images of 9 nm thick sections computational derived from tomograms of cuticulosomes from the basilar papillar of pigeons aged 8 days (n = 3 birds, n = 4 cells). (**b, e, h, k**) 3D-reconstructions of the cuticulosomes shown in panels a, d, g and j. Ferritin nanoparticles are shown in gold, vesicular structures in blue, and the apical membrane of the hair cell in green (**c, f, i, l**). Enlargements of boxes shown in b, e, h and k. In some, but not all, cases ferritin nanoparticles can be

*Figure 3 continued on next page*

*Figure 3 continued*

observed within vesicles (highlighted with a white arrowhead). In panels h and i a vesicle containing ferritin nanoparticles is visible and appears to be fusing with a larger membrane-bound cuticulosome. Note that in panels e and f the cuticulosome exhibits partial paracrystalline organisation with a linear arrangement of ferritin nanoparticles (highlighted with a white arrow). In panels j, k and l, the apical membrane of the cell is invaginated, indicative of an endocytotic event, with an accumulation of intracellular vesicles in its vicinity (red arrowheads). Scale bars show 200 nm in panels a, b, d, e, g, h, j and k and 100 nm in panels c, f, i and l.

DOI: https://doi.org/10.7554/eLife.29959.013

The following videos are available for figure 3:

**Figure 3—video 1.** Tomographic reconstruction of a developing cuticulosome.

DOI: https://doi.org/10.7554/eLife.29959.014

**Figure 3—video 2.** Tomographic reconstruction of a developing cuticulosome.

DOI: https://doi.org/10.7554/eLife.29959.015

**Figure 3—video 3.** Tomographic reconstruction of a developing cuticulosome.

DOI: https://doi.org/10.7554/eLife.29959.016

**Figure 3—video 4.** Tomographic reconstruction of a developing cuticulosome.

DOI: https://doi.org/10.7554/eLife.29959.017

the actin-rich meshwork that surrounds the cuticulosome is largely uniform (*Figure 2—figure supplement 1*). Taken together these data show that the maturation of the cuticulosome is associated with fusion of vesicles, some containing ferritin-like granules, resulting in an organelle that increases in size, density, and organisation with the passage of time.

## The cuticulosome contains ferritin

Our TEM analysis indicates that the cuticulosome is primarily composed of nanoparticles that resemble ferritin. To ascertain whether this is the case we undertook fluorescent histological staining with sera that target the heavy chain of ferritin, coupled with transmitted light microscopy that enables visualization of the cuticulosome (*Figure 4a–i*). This experiment revealed the presence of heavy chain ferritin throughout the cytoplasm of hair cells, with a notable punctate structure that coincided with the location of the cuticulosome in pigeons aged 1 day (n = 3 birds), 8 day (n = 2 birds) and 1 year (n = 3 birds) (*Figure 4d–i*). We confirmed this result by undertaking double staining of basilar papillae whole mounts with sera that bind to either actin or ferritin heavy chain (n = 3 birds per time-point). When visualized from a superior perspective we again observed a punctate ferritin positive structure in the cuticular plate of hair cells (*Figure 4j–l*). To assess the expression levels of ferritin heavy chain during development, we microdissected the distal region of the basilar papillar, extracted mRNA, generated cDNA and performed qPCR. This experiment revealed that the expression of heavy chain ferritin peaks at 16 days of age, but is consistently high at all developmental time points (*Figure 4—figure supplement 1*). It is well established that the dominant iron species in ferritin is the iron-oxide ferrihydrite, but it has also been reported to contain phases of magnetite ($Fe^{2+}$-$Fe^{3+}_2O_4$) and maghemite ($Fe_2O_3$) (*Cowley et al., 2000*; *Quintana et al., 2004*; *Gossuin et al., 2005*). We therefore performed Turnbull Blue staining (which labels $Fe^{2+}$) on sections from the basilar papilla and lagenar macula at hatching, 8 days, 16 days, 30 days and 1 year (n = 3 birds per time-point). As the staining was very light, indicative of low levels of $Fe^{2+}$, we post stained sections with the chromogen 3,3'-Diaminobenzidine (DAB) which is known to amplify the Turnbull Blue reaction (*Wang et al., 2002*). We observed a sharp increase in Turnbull blue staining in the basilar papilla and lagena from hatching to 30 days, mirroring the results of our Prussian Blue staining (*Figure 4—figure supplement 2*). These data indicate that the cuticulosome is composed of ferritin and that it contains $Fe^{2+}$ at low levels, consistent with the presence of multiple iron oxide species.

## Transcriptomic analysis of hair cells with cuticulosomes

To gain further insight into the molecular architecture of cuticulosome containing hair cells, we performed a transcriptomics experiment comparing RNA expression profiles of hair cells with and without cuticulosomes. To do so we adapted existing methods that enable ultra low input RNA sequencing, applying it to avian hair cells (*Picelli et al., 2014*). This method involved the dissection and isolation of the pigeon cochlear duct (without the lagena) (n = 9 birds, aged 1 year), followed by a light trypsinization to obtain single hair cells. Hair cells were then screened for the presence or

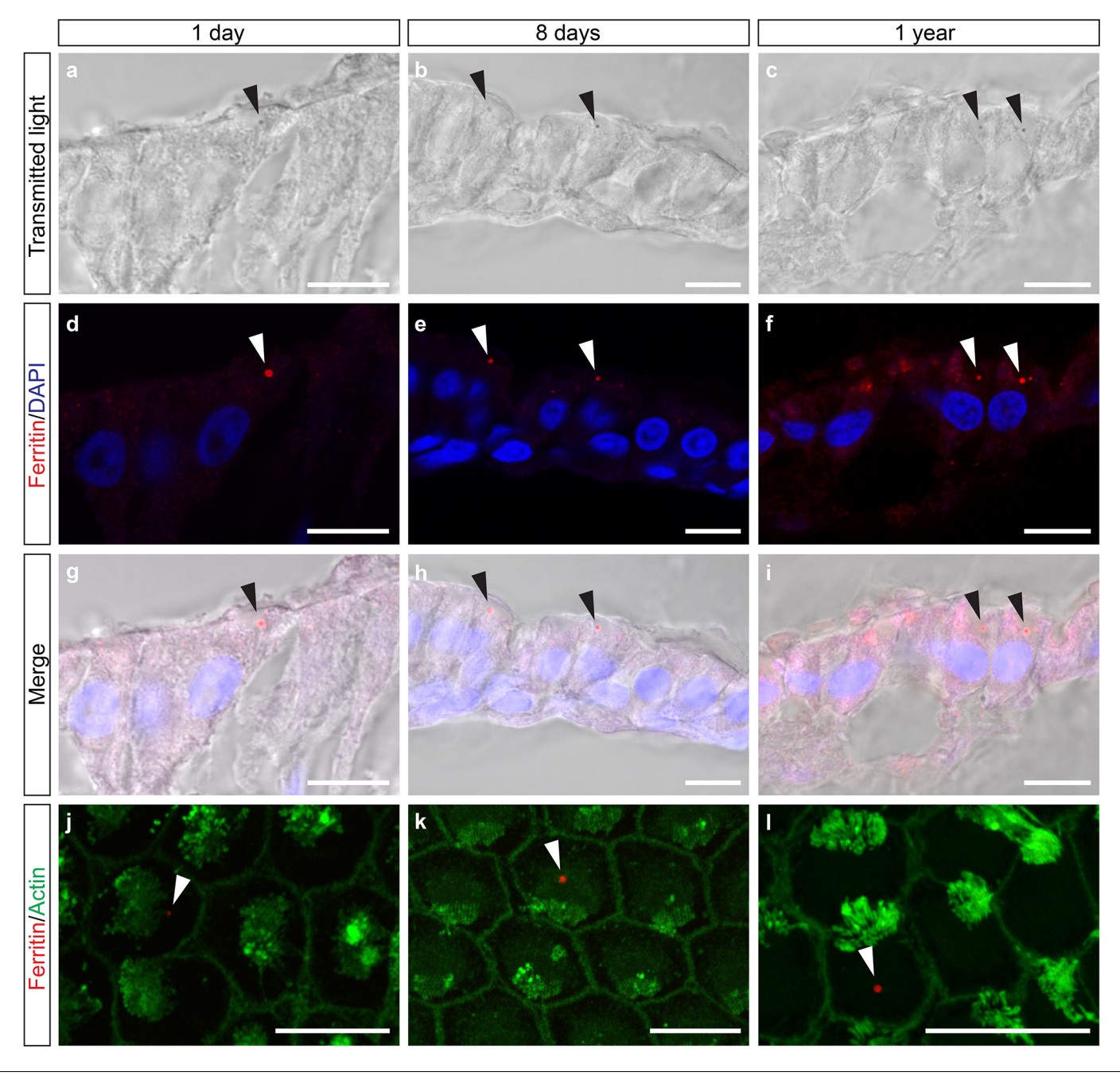

**Figure 4.** Heavy chain ferritin is a component of the cuticulosome. (a–c) Transmitted light images of sections of the basilar papilla of 1 day, 8 days and 1 year old pigeons. Cuticulosomes are highlighted with black arrows. (d–f) Confocal images of the same sections, showing immunoreactivity of cuticulosomes with sera against ferritin heavy chain. Punctate structures are highlighted with white arrows. (g–i) Merged images of transmitted light and immunostainings confirming that heavy chain ferritin staining co-localizes with cuticulosomes. (j–l) Confocal images of basilar papilla whole mounts from 1 day, 8 days and 1 year old pigeons stained with sera against ferritin and actin. The ferritin positive cuticulosomes, shown in red, are highlighted with white arrows and the actin rich stereocilia are shown in green. All scale bars show 10 μm. We have replicated this experiment three times.

DOI: https://doi.org/10.7554/eLife.29959.018

The following figure supplements are available for figure 4:

**Figure supplement 1.** Ferritin expression during development.
DOI: https://doi.org/10.7554/eLife.29959.019

**Figure supplement 2.** Quantitation of Turnbull blue positive hair cells.
DOI: https://doi.org/10.7554/eLife.29959.020

absence of cuticulosomes using light microscopy at 4°C. Hair cells with (n = 30 cells) or without cuticulosomes (n = 30 cells) were then picked with a micro-manipulator equipped with a fine glass-micropipette (*Figure 5a*). RNA was extracted and cDNA prepared, before next generation sequencing on an Illumina HiSeqV4 system. We chose a n = 9 for this experiment because it has previously been shown that a larger sample size increases the power for RNAseq experiments, however, in a paired experimental design this reaches a ceiling at n = 10 (*Ching et al., 2014*). We generated approximately 42 million reads per sample that were aligned with TopHat against the *Columba livia* genome (*Trapnell et al., 2009*) (*Shapiro et al., 2013*), and were subjected to FPKM estimation with Cufflinks (*Trapnell et al., 2010*, *Roberts et al., 2011*).

In total we identified 11,300 transcripts (mean FPKM > 1) with 10,277 (90.1%) having an annotated function (*Figure 5—source data 1*). To confirm the validity of this dataset we quantitated known hair cell markers such as otoferlin (mean FPKM > 1000), myosin 7A (mean FPKM > 35), protocadherin 15 (FPKM > 20), and cadherin 23 (mean FPKM > 7) (which comprise the tip link), and the putative mechanotransducer TMC2 (FPKM > 1000) (*Hasson et al., 1995*; *Roux et al., 2006*; *Kazmierczak et al., 2007*; *Pan et al., 2013*) (*Figure 5—figure supplement 1c*). In contrast the expression level of the support cell markers glial fibrillary acidic protein (GFAP) and SOX2, were below background (mean FPKM < 1) (*Rio et al., 2002*; *Hume et al., 2007*; *Togashi et al., 2011*) (*Figure 5—figure supplement 1c*). We plotted the twenty transcripts that were most highly expressed in cuticulosome positive and cuticulosome negative hair cells (*Figure 5—figure supplement 1a–b*). In both instances, house keeping genes (e.g. GAPDH, actin) as well as calcium binding proteins were dominant (calbindin, parvalbumin). Interestingly, the heavy chain of ferritin (FTH) was the second highest expressed transcript in both cell types (mean FPKM >11000), whereas the light chain of ferritin was present at very low levels (FPKM < 11) (*Figure 5—figure supplement 1a–b*).

Next, transcripts were subject to differential expression analysis with DESeq2 (*Anders and Huber, 2010*; *Love et al., 2014*) (*Figure 5b*). We found that 387 genes were upregulated (more than 3-fold increase) in cuticulosome positive cells, with 18 reaching significance (p<0.05) following correction for multiple testing (*Figure 5b–c* and *Figure 5—source datas 1* and *2*). In total 121 genes were downregulated (more than three fold reduction) in cuticulosome positive cells with seven reaching significance (p<0.05) (*Figure 5b* and *Figure 5—source datas 1* and *2*). We performed a gene ontology (GO) enrichment analysis of the 387 genes upregulated in cuticulosome positive cells, as the analysis of the significant hits alone did not identify any categories with enrichment (*Mi et al., 2013*) (*Figure 5c*). When analysing the cellular components enriched in cuticulosome positive cells we identified nine categories including: organelle (GO:0043226); membrane-bounded organelle (GO:0043227); and extracellular exosome (GO:0070062). A GO analysis of molecular function identified eight categories including: binding (GO:0005488); catalytic activity (GO:0003824) and metal ion binding (GO:0046872) (*Figure 5c*, see also *Figure 5—source datas 3–4*). Next we performed a manual annotation of all genes significantly up and down regulated in cuticulosome positive cells (*Figure 5—source data 2*). Interestingly, those genes significantly upregulated in cuticulosome positive cells included: RAB5B (which is involved in mediating early steps of endocytosis) (*Wilson and Wilson, 1992*); STXBP5L (a homologue of STXBP5 which regulates syntaxin mediated membrane fusion); and HEBP2 (which is a heme binding protein). Notably, RNF128 (a E3 ubiquitin ligase known to be involved in the trafficking of endosomes) and OSBPL2 (which belongs to a family of phosphatidylinositol 4-phosphate binding proteins that have been implicated in endosomal-endoplasmic reticulum tethering) were significantly down regulated (*Yamazaki et al., 2013*; *Dong et al., 2016*) (*Figure 5b* and *Figure 5—source data 2*). We confirmed the differential expression of RAB5B (n = 8 birds, p<0.05) and RNF128 (n = 8 birds, p<0.05) in cuticulosome positive and cuticulosome negative hair cells by qPCR (*Figure 5d*). Attempts to amplify transcripts expressed at low levels by qPCR (FPKM < 5), such as STXBP5L and CLIV_009848, were not successful. Taken together these data reveal that genes associated with vesicle trafficking, endocytosis, the formation of membrane bound organelles, and metal binding are differentially expressed in cuticulosome positive hair cells.

## Discussion

In this study we investigated the developmental, subcellular, and molecular architecture of cuticulosome positive hair cells. We report that when pigeons hatch only a few cuticulosomes are present, followed by a striking increase in their prevalence at 8 and 16 days of age, with the highest number

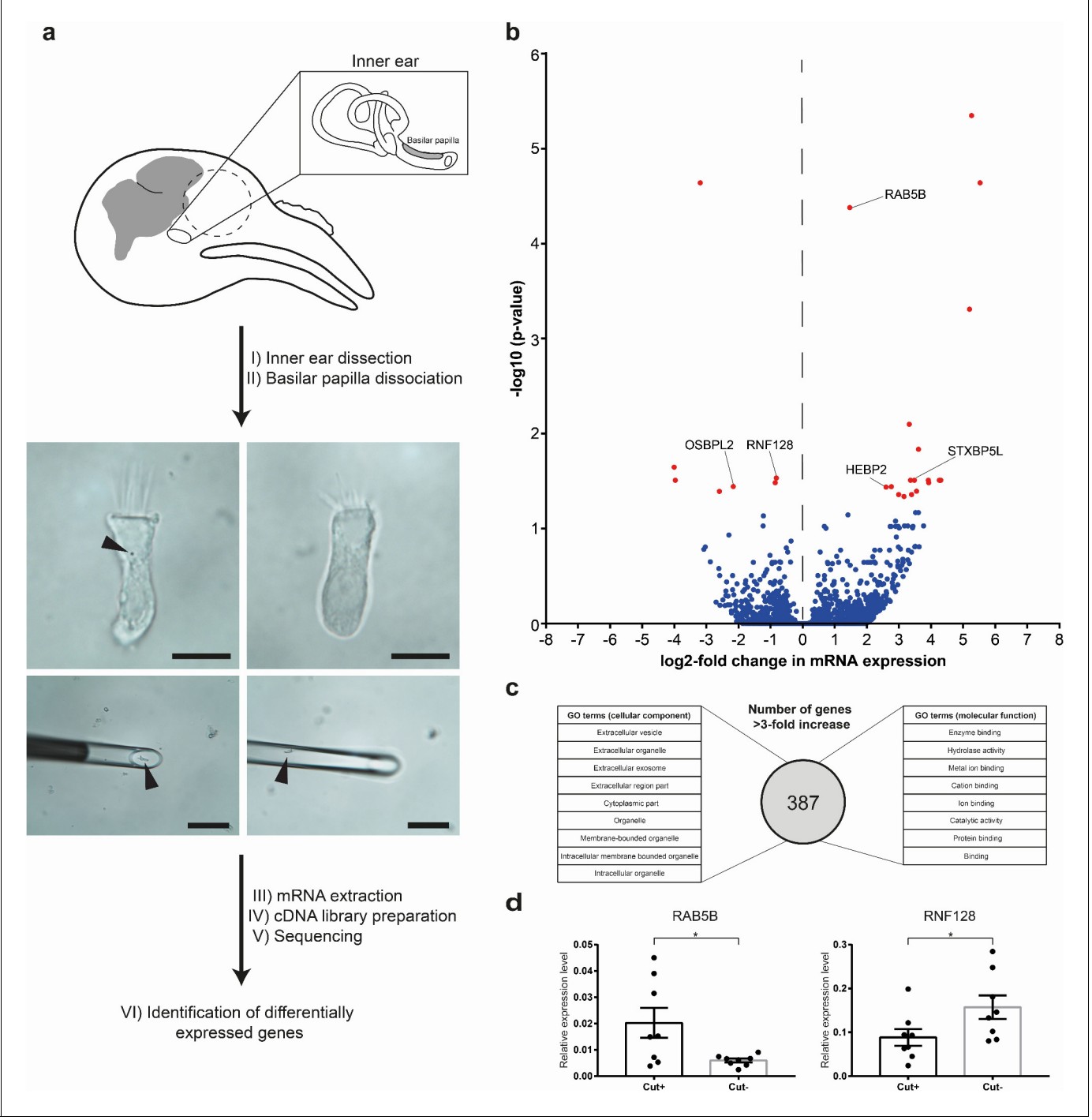

**Figure 5.** Transcriptomic analysis of hair cells with or without cuticulosomes. (**a**) Diagram showing the methodology employed for transcriptomic analysis. Following the dissection of the cochlear duct and removal of the lagena, the basilar papilla and surrounding tissue were subject to light trypsinization. Hair cells with (n = 30 hair cells, n = 9 birds), or without cuticulosomes (n = 30 hair cells, n = 9 birds), were then picked with a micromanipulator at 4°C. Following RNA extraction, and cDNA library preparation, next generation sequencing was performed, transcripts annotated and subject to differential gene expression analysis. (**b**) Volcano plot showing differential gene expression analysis between hair cells with and without cuticulosomes (n = 9 birds). The x-axis shows the log2-fold change in mRNA expression level when comparing hair cells with and without cuticulosomes and the y-axis shows corresponding adjusted P-values (-log10 scaled). Genes with a P- value ≤0.05, following correction for multiple testing, are highlighted in red. (**c**) Diagram representing the results of the gene ontology analysis for the 387 genes that were upregulated (more than 3-fold increase) in cuticulosome positive hair cells. (**d**) Quantitative real-time PCR results for RAB5B and RNF128 confirming differential expression of these transcripts in cuticulosome positive and cuticulosome negative cells (n = 8 birds). Gene expression levels of individual genes were calculated relative to

*Figure 5 continued on next page*

*Figure 5 continued*

the geometric mean of the control genes GAPDH and HPRT and plotted as mean ± SEM. (*p-value<0.05, paired one-tailed t-test). The scale bars in a represent 10 μm in the top panels and 100 μm in the bottom panels.

DOI: https://doi.org/10.7554/eLife.29959.021

The following source data and figure supplement are available for figure 5:

**Source data 1.** Transcripts Identified by RNA Sequencing.

DOI: https://doi.org/10.7554/eLife.29959.023

**Source data 2.** Significantly differentially expressed genes between hair cells with and without iron-rich organelles.

DOI: https://doi.org/10.7554/eLife.29959.024

**Source data 3.** GO enrichment analysis for 'cellular component'.

DOI: https://doi.org/10.7554/eLife.29959.025

**Source data 4.** GO enrichment analysis for 'molecular function'.

DOI: https://doi.org/10.7554/eLife.29959.026

**Figure supplement 1.** Analysis of RNA sequencing results.

DOI: https://doi.org/10.7554/eLife.29959.022

observable at 30 days. Our electron microscopy studies have further demonstrated that the cuticulosome is formed within the cuticular plate itself. The development of the cuticular plate, which is closely associated with the formation of stereocilia, begins at embryonic day 8 in chickens. At this time it is primarily occupied by vesicles, mitochondria, and lysosomes, before growing basally until embryonic day 14 when it resembles that present in the adult cochlea (*Tilney and DeRosier, 1986*). Post hatching the cuticular plate, with its web of actin filaments, is considered to be so dense that it excludes all organelles, even ribosomes (*Corwin and Warchol, 1991*). To enable the passage of vesicles to and from the apical membrane hair cells have a specialized zone close to the cell border that serves as a highway for subcellular structures that support membrane recycling (*Kachar et al., 1997*).

Our data indicate that the formation of the cuticulosome is initiated at the apical cellular border opposite the kinocilium. It appears that the cuticulosome is formed by the fusion of vesicles, some containing ferritin, that result in an organelle that increases in size and density (*Figure 6*). It appears that these vesicles are transported into the cuticular plate from the lateral boundary. As the density of the organelle increases, so does the likelihood that it develops a paracrystalline organization which results in alignment of the ferritin nanoparticles. The alignment of ferritin nanoparticles is uniform in some instances, but more often there are multiple domains within a cuticulosome that orient in different directions. Whether this organisation is facilitated by an active process, or is simply a consequence of an increase in density, is unclear. It should be noted, however, that iron-rich siderosomes that are associated with the formation of magnetite in chitons, rarely exhibit paracrystalline packing (*Shaw et al., 2009*). Intriguingly, our results also show that the position of the cuticulosome changes from an apical location at the edge of the cuticular plate to a medial, deeper position as it matures. This may reflect: (1) a post hatching expansion of the cuticular plate driven by actin polymerization along the apical periphery, resulting in a more centrally localized cuticulosome; or (2) a gradual remodeling of the cuticular plate, whereby the actin-rich meshwork is depolymerized by molecules such as cofilin. Mechanical forces generated by auditory stimuli, or motor proteins such as myosin, might then be responsible for the repositioning of the cuticulosome. This relocation does not appear to rely on microtubule dependent mechanisms, as we have not observed any evidence that the striated organelle associates with this organelle (*Vranceanu et al., 2012*).

Our results have also yielded insight into the molecular architecture of cuticulosome positive hair cells. Employing immunohistochemical methods we have shown that the cuticulosome contains the ferritin heavy chain protein. Moreover, given our RNA sequencing results, which have revealed that the ferritin light chain is expressed at very low levels, we would expect that the cuticulosome is dominated by the heavy chain isoform. Our transcriptomic studies have further implicated molecules associated with vesicular trafficking, endosomal sorting, extracellular exosomes and metal binding with cuticulosome maintenance. Of particular interest are RAB5B and RNF128. RNF128, which was significantly downregulated in cuticulosome containing hair cells, is a Ring finger E3 ubiquitin ligase. Drosophila and human family members of this protein class have been associated with the formation of enlarged Rab5 positive endosomes both in knockdown and overexpression studies

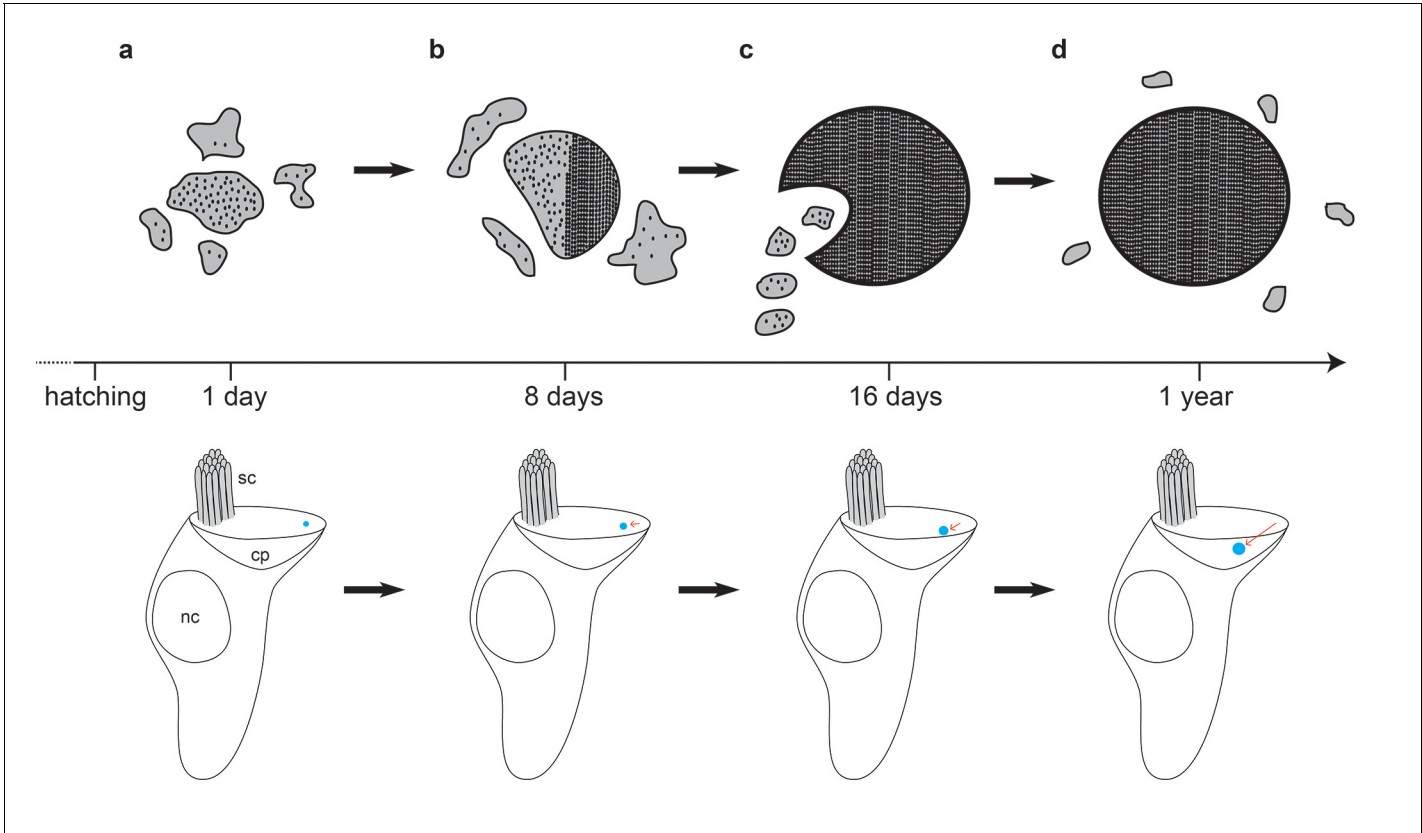

**Figure 6.** Model of cuticulosome development. (**a**) Cuticulosomes start to form shortly after hatching at the lateral apical edge of the cuticular plate, opposite the kinocilium. They initially resemble electron dense accumulations of amorphous shape with a diameter of approximately 300 nm. (**b**) Eight days after hatching the percentage of cuticulosome containing hair cells in the basilar papilla has increased 8-fold (20%) in comparison to 1 day old pigeons (2.5%). The size of the cuticulosome has increased to approximately 400 nm in diameter, with a notable increase in density. This increase in density is correlated with the development of paracrystalline arrays which result in the alignment of the ferritin nanoparticles. (**c**) Sixteen days after hatching cuticulosomes have reached their mature size (~580 nm), and their position within the cuticular plate has become more medial and basal. Distinct trails of vesicles from the lateral boundary of the cuticular plate are evident in some instances. (**d**) At one year of age cuticulosomes are mature. Approximately one quarter are still associated with vesicular structures and some exhibit complete paracrystalline organisation.
DOI: https://doi.org/10.7554/eLife.29959.027

(*Yamazaki et al., 2013*). Notably, cell lines that overexpress either wildtype Rab5 or a constitutively active form (rab5:Q79L) develop enlarged early endosomes (*Roberts et al., 1999*). It is therefore conceivable that the reduced expression levels of RNF128 and increased expression of RAB5B we observe in cuticulosome positive cells, licenses the formation of such a large membrane bound organelle within the cell.

The critical question that remains is: What is the function of the cuticulosome? And might it be associated with the magnetic sense? This is a tempting proposition given that pigeons begin to fly at approximately one month of age, a time when the cuticulosome is on the cusp of maturation (*Hazard, 1922*; *Nam and Lee, 2006*). However, this correlation should be viewed cautiously as there is conflicting behavioural data supporting the presence of a magnetic sense in young birds. Wiltschko and colleagues have reported that chickens aged 10 days can be conditioned to a magnetic anomaly (*Denzau et al., 2011*) and juvenile European Robins are known to possess a magnetic sense (*Muheim et al., 2002*), whereas pigeons aged 12 weeks are disorientated if they are transported in the dark (*Wiltschko and Wiltschko, 1981*). An experiment that assessed whether magnetic stimuli result in c-fos associated neuronal activation in pigeons of different ages would be valuable (*Wu and Dickman, 2011*).

The data presented in this manuscript strongly suggest that the formation of the cuticulosome is not the product of late-onset iron accumulation, and is more likely the result of a genetically

encoded developmental or homeostatic program. Could the cuticulosome therefore serve as a torque based magnetoreceptor that would exert a direct force on a mechanosensitive channel located on the apical membrane? This would seem unlikely. Firstly, based on our current understanding of the magnetic properties of ferritin, the cuticulosome lacks sufficient magnetic susceptibility in an Earth strength magnetic field to open a mechanosensitive channel, and, secondly, there seems to be no direct physical connection with the apical membrane that would mediate the transfer of a force (*Jandacka et al., 2015*). Indeed the cuticulosome appears to move deeper into the cuticular plate away from the apical membrane as it matures. An alternative mechanism proposed by Jandacka and colleagues is that the cuticulosome could act as an intracellular magnetic oscillator, that would induce an electrical potential across the cell membrane by auditory induced vibration of the basilar membrane (*Jandacka et al., 2015*). This hypothesis is problematic, however, as it requires the presence of an amplification mechanism to generate a voltage that exceeds thermal background. A third possibility is that the cuticulosome plays an indirect role in the magnetic sense by influencing the ionic concentration of the endolymph. A number of investigators have proposed that the detection of magnetic fields may rely on electromagnetic induction within the semicircular canals - the endolymph serving as a conductor (*Viguier, 1882*; *Jungerman and Rosenblum, 1980*). The cuticulosome, coupled with the vesicular machinery associated with it, might represent a means of regulating the conductivity of the endolymph by adjusting the concentration of cations, thereby tuning a magnetic sensor reliant on electromagnetic induction. Alternatively, the cuticulosome may serve as a precursor for a magnetite-based magnetoreceptor located elsewhere in the inner ear. Whether or not this proves to be the case, based on its striking subcellular localisation, elemental composition, and stereotypical early development, it seems unlikely that the presence of the cuticulosome is a mere coincidence.

## Materials and methods

### Animals

Birds were maintained on a 12:12 light-dark cycle at 25°C in a custom built aviary. All experiments were performed in accordance with an existing ethical framework (GZ: 214635/2015/20) granted by the City of Vienna (Magistratsabteilung 58).

### Histological studies

The cochlear ducts from pigeons (*Columba livia*) of different ages were dissected with ceramic-coated tools and drop fixed in 4% phosphate buffered formaldehyde (PFA) (pH 7.4). After dehydration in a series of ethanol concentrations, the cochlear ducts were embedded in paraffin and sectioned on a rotary microtome (10 μm) with Dura Edge High Profile ceramic-coated blades (Ted Pella Inc. Redding, CA, USA, BLM00103P). For Prussian blue (PB) stainings, slides were incubated for 20 min in 5% potassium hexacyanoferrate (II) (Sigma, Germany, P9387) with 10% HCl. The slides were washed 3 times for 5 min in PBS and then counterstained with Nuclear Fast Red (Sigma, 60700) for 1 min, followed by another washing step, dehydration and cover-slipping. All hair cells and PB-positive hair cells on every fourth section were counted and multiplied by a factor of 4 to obtain estimates of total hair cell number and total PB-positive hair cell number in each cochlear duct. For whole mount experiments, the tegmentum vasculosum and the tectorial membrane were removed, the cochlear ducts were then fixed in 4% PFA (pH 7.4), washed in PBS, incubated in 5% potassium hexacyanoferrate (Sigma, P9387) with 10% HCl for 20 min, followed by another wash in PBS and counterstained in Nuclear Fast Red for 1 min. The whole mounts were embedded in 87% glycerol (BioChemica, UK, A0970) and coverslipped. For Turnbull blue (TB) stainings, 5–10 sections per bird were incubated for 20 min in 5% potassium hexacyanoferrate (III) (Sigma, P4066) with 10% HCl (n = 3 per timepoint). The slides were washed 3 times for 5 min in PBS after which endogenous peroxidase was blocked by 30 min of incubation in 0.3% $H_2O_2$ in methanol. After three further PBS washes, the TB staining was intensified by incubating with 3,3' Diaminobenzidine (Carl Roth, Germany, CN75.2). Another three washes with PBS were followed by counterstaining with Nuclear Fast Red (Sigma, 60700) for one minute, another washing step, dehydration and coverslipping. All hair cells and TB-positive hair cells on at least five sections per bird and region (lagena and basillar papilla) were counted.

## Ultrastructural studies

Cochlear ducts from pigeons of different ages were dissected before fixing overnight in 2.5% glutar-aldehyde supplemented with 2% paraformaldehyde (PFA) in Soerensen buffer (pH 7.4). Tissues were then incubated in osmium tetroxide (2%, pH 7.4, Soerensen buffer) for 1 hr, dehydrated in increasing ethanol solutions, embedded in epoxy resin and polymerized at 60°C for 48 hr. Ultra-thin (70–120 nm) sections were prepared and analyzed with a Morgagni 268D transmission electron microscope operated at 80kV (FEI, Hillsboro, Oregon, USA). The size of cuticulosomes was measured on electron micrographs of cuticulosomes using the ImageJ software (http://imagej.net). We defined a cuticulo-some as being in formation if its size was less than 250 nm, and/or the periphery of the cuticulosome was amorphous. Density plots were generated by drawing lines (200 nm each) radially from the edge of cuticulosomes in 45° increments (*Figure 2—figure supplement 1*). The density along these lines was plotted in 0.372 nm increments using the Imagej's plot profile function (http://imagej.net). The mean intensity of every increment was calculated for eight cuticulosomes individually (n = 3 birds aged 8 days).

## Tomography and modeling

Epon embedded sections (200–250 nm) were collected on slot grids, poststained with 2% aqueous uranyl-acetate followed by Reynold's lead citrate. For image alignment during tomogram reconstruction 10 nm gold beads Aurion (Wageningen, 410.011) were added as fiducial markers. Room temperature tilt series were acquired at a Tecnai G2 20 microscope (FEI) operated at 200kV and equipped with an Eagle 4 k HS CCD camera (FEI). Dual axis tilt series were collected using the SerialEM acquisition software with a tilt range from −60° to +60° with one degree increments (*Mastronarde, 2005*). Tomogram reconstruction and modelling were done using the IMOD software (*Kremer et al., 1996*). To eliminate noise in the isosurfaced model, binning of the original data, filtering for a minimum size of structures, and masking with the cuticulosome perimeter object were used.

## Immunohistochemical studies

The cochlear ducts of pigeons were dissected as described above and fixed in 4% phosphate buffered formaldehyde (PFA) (pH 7.4) for 10 min. After fixation, the tissue was dehydrated in 30% Sucrose/PBS overnight, embedded in Neg-50 Frozen Section Medium (Thermo Fisher, Waltham MA, USA, 6502) and sectioned (12 μm) on a cryostat (Thermo Fisher, MICROM HM 560). Slides were dried for 3 hr at room temperature, washed three times in PBS for 5 min followed by incubation in antigen unmasking solution (Vectorlabs, Berlingham, CA, USA, H-3301) in a waterbath that was heated up to 90°C. Slides were cooled down at room temperature for 30 min afterwards, washed three times in PBS for 5 min and incubated with a ferritin heavy chain primary antibody (Santa Cruz, Dallas, Texas, USA, sc-14416) over night at 4°C at a concentration of 1:500 in 0.3% Triton/PBS (pH 7.4) supplemented with 2% donkey serum. Slides were washed three times in PBS for 5 min, incubated with donkey anti-goat Alexa Fluor-568 secondary antibody (Thermo Fisher, A-11057) at a concentration of 1:500 for 1 hr at 4°C. This was followed by another washing step, coverslipping and imaging using a confocal microscope (Zeiss, Germany, LSM 700). Whole mounts were fixed in 4% PFA for 10 min, washed in PBS following antigen retrieval as described above and incubated with primary antibodies for heavy chain ferritin (1:500) and actin (Millipore, Burlington, MA, USA, MAB1501, 1:2000) overnight at 4°C in 0.3% Triton/PBS (pH 7.4). This was followed by incubation with secondary antibodies donkey anti-goat Alexa Fluor-568 (Thermo Fisher, A-11057) and donkey anti-mouse Alexa Fluor-488 (Thermo Fisher, A-21202) at a concentration of 1:500 for 1 hr at 4°C, before washing and coverslipping. Z-stacks were acquired using a confocal microscope and z-projections with maximum intensity were generated using the ImageJ software.

## cDNA library preparation for RNA sequencing

The cochlear ducts from pigeons (1 year of age) were dissected with titanium tools and transferred to Bird Ringer solution (145 mM NaCl, 5 mM KCl, 1 mM CaCl$_2$, 1 mM MgCl$_2$, 5 mM Glucose, 20 mM HEPES, DEPC treated Mono Q H$_2$O, pH 7.4). The lagenar maculae were removed from the cochlear ducts with a ceramic scalpel blade. The remaining parts of the cochleae were incubated in 0.05% Trypsin:EDTA (GIBCO, Carlsbad CA, USA, 15400–054) at 37°C for 5 min and cut into small pieces.

Trypsinization was stopped by adding trypsin inhibitor (0.5 g/100 ml, Sigma Aldrich, T6522-25MG) and recombinant RNAse inhibitor was added to prevent RNA degradation (Takara, Japan, 2313A). Hair cells were screened and isolated by employing a modified, inverted microscope (Zeiss, Axiovert 135), using anmicromanipulator (TransferMan, Eppendorf, Germany, NK2) equipped with glass micropipettes (BioMedical Instruments, Clinton Township, MI, USA, BM100T-15). RNA extractions of 30 isolated hair cells with and without cuticulosomes from nine birds respectively (18 samples in total) were performed using the TRIZOL/Chloroform method and the RNA Clean and Concentrator-5 Kit (Zymo Research, Irvine , CA , USA, R1015). A total of 21 cDNA libraries (18 samples plus three negative controls) were prepared employing a full length RNA-seq from single cells protocol (*Picelli et al., 2014*). Briefly, this method involved a preamplification of reverse transcribed mRNA using a template switching oligo followed by fragmentation and adaptor ligation using a hyperactive derivative of the Tn5 transposase from the Nextera XT DNA Sample Preparation Kit (Illumina, San Diego, CA, USA, FC-131–1096). Adapter-ligated DNA fragments were amplified by PCR using the Nextera XT DNA Sample Preparation Kit and the Nextera XT 24-index Kit (Illumina, FC-131–1001). The quality of the cDNA in each sample was analyzed using the Agilent High Sensitivity DNA Kit (Agilent Technologies, Santa Clara, CA, USA, 5067–4626) and the Agilent 2100 Bioanalyzer (Agilent Technologies, G2940CA) to ensure that only libraries with high quality were selected for sequencing.

## Sequencing and bioinformatic analysis

cDNA libraries were pooled in equal molarity for multiplexing and sequencing was performed on an Illumina HiSeqV4 system with 50 base pair single-end reads using the True-Seq dual-index sequencing primers for single read runs (Illumina, FC-121–1003). The sequencing data was analyzed using an adapted published protocol (*Berger et al., 2012*). Briefly, the reads were aligned with TopHat (v1.4.1) (*Trapnell et al., 2009*) against the *Columba livia* genome (provided by M.D. Shapiro, University of Utah). A maximum of 6 miss matches and introns between 20–150000 bp were allowed. Maximum multihits were set to one and InDels and Microexon-search was permitted. Furthermore, a gene model was implemented as GTF (Shapiro). Aligned reads were subjected to FPKM estimation with Cufflinks (v1.3.0) (Trapnell, Williams et al. 2010, Roberts, Trapnell et al. 2011) and bias detection and correction was performed. Only fragments compatible with annotation were accepted and counted towards the number of mapped hits used in the FPKM denominator. The aligned reads were counted with HTSeq (0.6.1p1) and the transcripts were subjected to differential expression analysis with DESeq2 that incorporates a Benjamini and Hochberg procedure for multiple testing (*Anders and Huber, 2010*; *Love et al., 2014*). Transcripts with an FPKM >1 were considered to be expressed. Since cuticulosome and non-cuticulosome cells were collected from the same birds, this was considered via blocking in the differential expression analysis. We define a biological replicate as a separate biological entity, in contrast to a technical replicate which simply involves multiple measurements of the same sample. All transcriptomic analysis involved biological replicates.

## Real time quantitative PCR

To assess the levels of ferritin heavy chain microdissections of the cochlear duct were performed on the distal region of the basilar papillar isolated from birds at hatching, 8 days, 16 days, 30 days and 1 year (n = 3 birds per timepoint). Total RNA from hair cell epithelia was extracted using the RNeasy Mini Kit (Qiagen, Netherlands, 74104) and cDNA was prepared using the QuantiTect Reverse Transcription Kit (Qiagen, 205313). Intron-spanning qPCR primers for each gene of interest were designed using the Primer3Plus program (http://primer3plus.com/cgi-bin/dev/primer3plus). The efficiency of each primer pair was assessed and only primers with efficiency between 90% and 100% were selected for the experiments. Reactions were performed in technical triplicates together with GAPDH and HPRT as control genes and no template controls on a Bio-Rad CFX Connect cycler (Bio-Rad, Hercules, CA, USA, 1855201) using the SsoAdvance Universal SYBR Green Supermix (Bio-Rad, 1725271). Gene expression levels of individual genes were calculated relative to the geometric mean of GAPDH and HPRT. To confirm our transcriptomic results qPCR was performed on cDNA libraries prepared from cuticulosome positive and cuticulosome negative cells as described above (n = 8 birds). Statistical analysis was performed using paired one-tailed t-tests followed by Bonferroni correction for multiple comparison.

The following primers were used: pHPRT_F AAATTTGTCGTGGGATACGC; pHPRT_R TACTTC TGCTTCCCCGTCTC; pGAPDH_F TCAATGGGAAACTTACTGGAATGG; pGAPDH_R TCTTAATG TCATCATACTTGGCTGG; pRab5b_F ATGGTGGAGTACGAGGAGG; pRab5b_R AAGAGATCG TTGACGTTCATGG; pRnf128_F ATGTTTGGTATATACGTGACAGCC; pRnf128_R TCTAGAACCA TCCAGAAACCAAAC; pFerHeavy_F TACGTGTATCTCAGCATGTCCTAC; pFerHeavy_R TACGTGTA TCTCAGCATGTCCTAC.

## Acknowledgements

DAK is supported by the European Research Council (ERC, 336725) and the FWF (Y726). Simon Nimpf is a recipient of a DOC Fellowship from the Austrian Academy of Sciences. We wish to thank Boehringer Ingelheim who fund basic scientific research at the Research Institute of Molecular Pathology (IMP). We are indebted to the excellent support facilities at the IMP and VBCF including: histology, bio-optics, electron microscopy, and bioinformatics.

## Additional information

### Funding

| Funder | Grant reference number | Author |
| --- | --- | --- |
| Austrian Science Fund | Y726 | Thomas R Burkard |
| Horizon 2020 Framework Programme | 336724 | David A Keays |

The funders had no role in study design, data collection and interpretation, or the decision to submit the work for publication.

### Author contributions

Simon Nimpf, Data curation, Formal analysis, Investigation, Methodology, Writing—original draft, Writing—review and editing; Erich Pascal Malkemper, Conceptualization, Methodology, Writing—review and editing; Mattias Lauwers, Data curation, Formal analysis, Investigation, Methodology; Lyubov Ushakova, Gregory Nordmann, Data curation, Investigation; Andrea Wenninger-Weinzierl, Data curation, Methodology; Thomas R Burkard, Software, Formal analysis; Sonja Jacob, Data curation, Formal analysis; Thomas Heuser, Data curation, Formal analysis, Writing—review and editing; Guenter P Resch, Data curation, Formal analysis, Investigation, Writing—review and editing; David A Keays, Conceptualization, Formal analysis, Supervision, Funding acquisition, Writing—original draft, Project administration, Writing—review and editing

### Author ORCIDs

Simon Nimpf http://orcid.org/0000-0001-6522-6172
David A Keays http://orcid.org/0000-0002-8343-8002

### Ethics

Animal experimentation: All experiments were conducted in accordance with an existing ethical framework GZ: 214635/2015/20 granted by the City of Vienna (Magistratsabteilung 58).

### Decision letter and Author response

Decision letter https://doi.org/10.7554/eLife.29959.031
Author response https://doi.org/10.7554/eLife.29959.032

## Additional files

### Supplementary files

• Transparent reporting form
DOI: https://doi.org/10.7554/eLife.29959.028

## Major datasets

The following dataset was generated:

| Author(s) | Year | Dataset title | Dataset URL | Database, license, and accessibility information |
|---|---|---|---|---|
| Burkard TR, Keays DA, Nimpf SA | 2017 | Transcriptomic analysis of hair cells from the pigeon Columba livia | https://www.ncbi.nlm.nih.gov/geo/query/acc.cgi?acc=GSE100823 | Publicly available at the NCBI Gene Expression Omnibus (accession no: GSE100823) |

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
