## [Decision Letter]

Thank you for submitting your manuscript "Subcellular analysis of pigeon hair cells implicates vesicular trafficking in cuticulosome formation and maintenance" to *eLife*. Three experts reviewed your manuscript, and their assessments, together with my own (Jeremy Nathans, Reviewing Editor), form the basis of this letter. The handling Senior Editor was Andrew King. As you will see, all of the reviewers were impressed with the importance and novelty of your work.

I am including the three reviews at the end of this letter, as there are a variety of specific and useful suggestions in them. We appreciate that the reviewers' comments cover a broad range of suggestions for improving the manuscript. Given these disparate concerns, we feel it would be best for you to evaluate the suggestions and write back to us with a prioritized list of experiments that could be completed in a reasonable period of time. We will share your plans with the reviewers and then respond with advice on how best to proceed.

Reviewer #1:

The cuticulosome is a newly discovered, orphan organelle in search of a function and a developmental understanding. In this manuscript, Nimpf et al. describe the distribution and formation of ferritin-containing cuticulosomes in hair cells from the basilar papilla of pigeons with high-resolution transmission electron micrographs taken at different developmental stages and observe that the cuticulosomes are surrounded by multiple vesicles with or without ferritin. In addition, single cell transcriptome analysis conducted in this manuscript reveals the potential importance of vesicle fusion in formation of cuticulosomes. And it also provides a rich resource in the form of a gene-candidate list that could now be used to understand the formation or the function of cuticulosomes in the future. This is a well-written and interesting manuscript, which could be improved with the below suggestions, for suitability in *eLife*.

Major concerns

1) The models put forth of cuticulosome development are interesting. I feel that to make the study more complete the author should also explore the cuticular plate actin structure immediately around the cuticulosome. How is the actin altered by the vesicles? Are there actin-free tunnels (or pockets) in which these vesicles may travel (or reside)? The answers to these questions should be addressed with tomography and added to Figure 3. This would help answer the developmental question they are addressing. This should not be too difficult since they likely have the data but would need to rework it. In addition, did the authors observe any structural differences in the cuticular plates in hair cells with and without the cuticulosome?

2) Can the authors localize RAB5B to the cuticulosome or any proteins from their list (preferred) that are differentially expressed? If not, can the authors perform in situ hybridization to show differential expression. The data needs validation. As a last resort, can they do single cell RT-PCR?

3) Why is there a very wide range of sizes of cuticulosomes in one-year-old pigeons? Since birds can regenerate hair cells, could this be that some of the cuticulosomes measured are in newly born hair cells? Or is there a big variation of cuticulosomes' sizes in a one-year-old pigeon. If so, can the authors further explain the criteria of selecting a single dissociated hair cell?

4) If, as the authors proposed, vesicles surrounding the developing cuticulosome help the cuticulosome to grow in size, is fusion between the vesicle and the cuticulosome ever observed by electron microscopy?

5) Figure 2—figure supplement 1 is missing.

6) When is the onset of hearing and flying ability in pigeons? Additional information on the stages of hair cell and ear development represented by the different ages chosen for analysis would be helpful in interpreting the results. Could the organized ferritin particle be used as a marker for maturation of the cuticulosome?

Reviewer #2:

This manuscript is a developmental and transcriptomic characterization of the pigeon hair cells that contain cuticulosomes, an organelle that was originally described by the Keays Lab. The current manuscript suggests that this organelle is tightly regulated and therefore must have an important function. However, what that function might be is very speculative.

The authors have performed a morphological analysis of this organelle using Prussian blue (PB) staining across post hatching developmental stages. Their novel finding in this section is that cuticulosomes form rapidly, increase in number through development and their localization changes from a lateral-apical to a medio-basal position as they mature. The characterization of hair cells using TEM reveals that hair cells from recently hatched animals contain cuticulosomes that are "diffuse" and around 16 days after hatching they become larger and paracrystalline in structure.

Q1) Is it possible that the observed change in size and structure is due to the TEM sample preparation and for some reason younger hair cell cuticulosomes are more susceptible to degradation?

Next, the authors demonstrate that the cuticulosomes are composed of nanoparticles of ferritin by performing ferritin heavy chain IHC.

Q2) Although the ferritin heavy chain-rich puncta resemble the cuticulosomes there is no quantification to demonstrate that these structures develop similar to PB-positive cuticulosomes and hence they are likely the same organelle.

The transcriptomic analysis of hair cells with and without cuticulosomes is done with 1-year-old animals. One of the most highly expressed transcripts is the ferritin heavy chain in cuticulosome positive and negative hair cells.

Q3) The ferritin IHC data in Figure 4 seem to indicate that there is diffuse heavy chain expression in all cells in 1-year-old animals. However, this background staining is missing in newly hatched birds. Since this is a developmental study that impinges on the presence of ferritin nanoparticles in cuticulosomes, it is critical to know the profile of the transcriptomic information from newly hatched birds, as well. At the very least, expression of ferritin heavy chain transcript across developmental stages should be confirmed by either in situ hybridization or single cell quantitative RT-PCR.

Differential expression analysis of the transcriptomic data reveals that "genes associated with vesicle trafficking, endocytosis, the formation of membrane bound organelles, and metal binding are differentially expressed in cuticulosome positive hair cells."

Q4) It is necessary to confirm at least some of these gene expression profiles with in situ hybridization.

Q5) In their abstract the authors state that "These data suggest that the cuticulosome and the associated molecular machinery regulate the concentration of Fe3+ within the labyrinth of the inner ear, which might indirectly tune a magnetic sensor that relies on electromagnetic induction." However, in their previous paper they have stated that "There is no evidence that sensory neurons play a role in iron homeostasis, and it is not clear why a role in iron storage would necessitate the conspicuous positioning of a single organelle in the cuticular plate." In addition, it is not clear why these structures would need to be composed of ferritin particles that are arranged in a paracrystalline array, heavily regulated developmentally in size/position and appear only in a percent of cells to perform the proposed homeostatic function.

Reviewer #3:

The manuscript by Keays and colleagues is the first study that investigates the formation and trafficking of cuticulosomes in pigeon hair cells during early development after hatching. The data revealed in this detailed manuscript are important for understanding the development of magnetoreception in young pigeons. However, a few questions should be addressed before the manuscript is suitable for publication:

1) One of the main concerns is the low-number animal cohorts (3-6) and few (e.g. 3) repetitions of each experiment. Was the power analysis ever performed, or what was the rationale for cohort size selection? (There's only one RNASeq experiment for which authors discuss their sample size selection.)

2) Subection “Cuticulosomes form rapidly after birds hatch”, percent averages of cuticulosome positive cells are provided with no standard deviation/error. Given the small cohort sizes it would be valuable to see the deviations.

3) As authors indicate that the ferritin clusters cuticulosomes may be contributing to magnetoreception, it would be interesting to identify what is the iron content of these structures – that is spacing between individual iron atoms.

4) Following up on the previous point, the magnetic properties of the cuticulosome are of great interest. Have the authors performed vibrating sample or SQUID magnetometry on these structures? Before magnetoreception can be discussed, the structures need to be proven magnetic or conductive (if the inductive mechanism is considered).

5) The authors suggest that the cuticulosome and the associated molecular machinery regulate the concentration of Fe3+ within the labyrinth of the inner ear. However, whether the concentration of Fe3+ changes during development is still unknown. The correlation between Fe3+ concentration in the labyrinth and the expression of cuticulosomes is also not clear. The authors need to provide stronger evidence or supporting references to show the relation between Fe3+ concentration in the labyrinth and the number of cuticulosomes in hair cells.

6) It is surprising that the number of cuticulosomes is the highest in 30-day old pigeons. The authors, however, do not show the localization or the size distribution of cuticulosomes in 30-day old pigeons as compared to 1-year old pigeons. The authors should demonstrate whether the cuticulosomes are fully developed in 30-day old pigeons.

7) In Figure 4, panels J–L would be better used by showing the immunostaining for Ferritin/Actin at 1 day, 8 days and 1 year for consistency with panels A–I. Showing 3 separate stains at 1 year is redundant (obviously multiple staining experiments must be performed for quantification of expression).

8) According to the results presented in this manuscript, cuticulosomes develop within a few days after hatching, and the number of cuticulosomes in the basilar papilla peaks in 30-day old pigeons. This is at odds with prior work that demonstrated that young pigeons (<12 weeks old) are incapable of perceiving magnetic field in the dark (Wiltschko and Wiltschko, 1981). The authors should discuss this in their manuscript.

References: Wiltschko, W. and Wiltschko, R., (1981) Disorientation of inexperienced young pigeons after transportation in total darkness. Nature 291: 433 – 434.

---

## [Author Response]

Reviewer #1:The cuticulosome is a newly discovered, orphan organelle in search of a function and a developmental understanding. In this manuscript, Nimpf et al. describe the distribution and formation of ferritin-containing cuticulosomes in hair cells from the basilar papilla of pigeons with high-resolution transmission electron micrographs taken at different developmental stages and observe that the cuticulosomes are surrounded by multiple vesicles with or without ferritin. In addition, single cell transcriptome analysis conducted in this manuscript reveals the potential importance of vesicle fusion in formation of cuticulosomes. And it also provides a rich resource in the form of a gene-candidate list that could now be used to understand the formation or the function of cuticulosomes in the future. This is a well-written and interesting manuscript, which could be improved with the below suggestions, for suitability in eLife.

We thank the reviewer for his/her kind remarks and constructive criticism.

Major concerns1) The models put forth of cuticulosome development are interesting. I feel that to make the study more complete the author should also explore the cuticular plate actin structure immediately around the cuticulosome. How is the actin altered by the vesicles? Are there actin-free tunnels (or pockets) in which these vesicles may travel (or reside)? The answers to these questions should be addressed with tomography and added to Figure 3. This would help answer the developmental question they are addressing. This should not be too difficult since they likely have the data but would need to rework it. In addition, did the authors observe any structural differences in the cuticular plates in hair cells with and without the cuticulosome?

The reviewer has raised an interesting issue regarding the architecture of the actin rich meshwork surrounding the cuticulosome. Specifically, the reviewer has asked whether there are any actin-free tunnels or pockets. To investigate this we have examined high-resolution electron micrographs of developing cuticulosomes (8 days post hatching). This has not revealed any obvious tunnels (or pockets) in which vesicles travel. To display this data quantitatively we measured the intensity of osmium staining along 200nm lines that extend radially from the edge of cuticulosomes (n= 8 lines per cuticulosome, n=8 cuticulosomes, n=3 birds). This result suggests that there are no "vesicle highways", but rather vesicles squeeze through the actin meshwork which presumably involves some remodeling of the structure. These data are shown in Figure 2—figure supplement 1 and the text has been updated accordingly (subsection “The subcellular architecture of cuticulosomes during development”).

2) Can the authors localize RAB5B to the cuticulosome or any proteins from their list (preferred) that are differentially expressed? If not, can the authors perform in situ hybridization to show differential expression. The data needs validation. As a last resort, can they do single cell RT-PCR?

We have found that few commercially available antibodies work well on pigeon tissue (we have tried RAB5, LAMP, RAB7). We have likewise had limited success employing *in situ* hybridization on the pigeon adult inner ear. Nevertheless, we agree with the reviewer that validation of our transcriptomic data is important. We have therefore undertaken qPCR analysis of our best candidates RAB5B and RNF128 on cuticulosome positive and negative cells (n=8 birds). These data which are now shown in Figure 5, confirm that RAB5B is enriched in cuticulosome positive cells and RNF128 is enriched in cuticulosome negative cells. Our attempts to validate additional candidates (such as STXBP5L and CLIV_009848) were unsuccessful as they are expressed at low levels (FPKM<5) and we were not able to amplify them by qPCR (in either cuticulosome positive or negative cells). The text has been updated accordingly, subsection “Transcriptomic analysis of hair cells with cuticulosomes”.

3) Why is there a very wide range of sizes of cuticulosomes in one-year-old pigeons? Since birds can regenerate hair cells, could this be that some of the cuticulosomes measured are in newly born hair cells? Or is there a big variation of cuticulosomes' sizes in a one-year-old pigeon.

The average size of cuticulosomes in 1 year old birds is 545 ± 31.4 nm (n = 18 cuticulosomes), which is similar to that in 30 day old birds (514.3 ± 31.4 nm (n = 17 cuticulosomes). It does seem that the variation in cuticulosome size is particularly pronounced at 1 year of age. However, as the reviewer suggests the smaller cuticulosomes at this time could reflect hair cell regeneration which is known to occur in Avian species. Those hair cells with smaller cuticulosomes may represent newly born hair cells. Indeed, this could account for the 7% of cuticulosomes that we observe at 1 year of age that appear to be in the process of formation (See Figure 2—source data 1).

If so, can the authors further explain the criteria of selecting a single dissociated hair cell?

When selecting single hair cells for the transcriptomic analysis, a visual inspection of individual hair cells was made by moving through the focal plane in the Z-direction. Those hair cells with a round opaque structure within the cuticular plate were determined to be cuticulosome positive.

4) If, as the authors proposed, vesicles surrounding the developing cuticulosome help the cuticulosome to grow in size, is fusion between the vesicle and the cuticulosome ever observed by electron microscopy?

In our electron tomograms we observed vesicular structures that looked like they were about to fuse with the cuticulosome (see Figure 3) but none that were in the process of fusion.

5) Figure 2—figure supplement 1 is missing.

This has now been rectified and the relevant table inserted. See Figure 2—source data 1.

6) When is the onset of hearing and flying ability in pigeons? Additional information on the stages of hair cell and ear development represented by the different ages chosen for analysis would be helpful in interpreting the results.

Flying in pigeons occurs at approximately one month of age. It is therefore tempting to correlate this behavioural milestone with the formation of the cuticulosome. However, we recommend caution on this front as it is unclear whether young pigeons have a magnetic sense.

We now state in the Discussion section:

"The critical question that remains is: What is the function of the cuticulosome? And might it be associated with the magnetic sense? This is a tempting proposition given that pigeons begin to fly at approximately one month of age, a time when the cuticulosome is on the cusp of maturation (Hazard, 1922, Nam and Lee, 2006). However, this correlation should be viewed cautiously as there is conflicting behavioural data supporting the presence of a magnetic sense in young birds. Wiltschko and colleagues have reported that chickens aged 10 days can be conditioned to a magnetic anomaly (Denzau, Kuriakose et al., 2011) and juvenile European Robins are known to possess a magnetic sense (Muheim, Backman et al., 2002), whereas pigeons aged 12 weeks are disorientated if they are transported in the dark (Wiltschko, 1981). An experiment that assessed whether magnetic stimuli results in c-fos associated neuronal activation in pigeons of different ages would be valuable (Wu and Dickman, 2011)."

We are not aware of any specific study in pigeons that have assessed their hearing at different ages, however, for chickens it is known that the tonotopic map of the cochlear develops during embryogenesis, and at 21 days of age the auditory system is mature structurally and functionally (Jones and Jones, 1995).

Could the organized ferritin particle be used as a marker for maturation of the cuticulosome?

As only 16% of cuticulosomes are ordered in a paracrystalline array in the adult (and this ordering is often partial), we do not think this particular attribute is best for defining whether or not a cuticulosome is mature.

Reviewer #2:This manuscript is a developmental and transcriptomic characterization of the pigeon hair cells that contain cuticulosomes, an organelle that was originally described by the Keays Lab. The current manuscript suggests that this organelle is tightly regulated and therefore must have an important function. However, what that function might be is very speculative.The authors have performed a morphological analysis of this organelle using Prussian blue (PB) staining across post hatching developmental stages. Their novel finding in this section is that cuticulosomes form rapidly, increase in number through development and their localization changes from a lateral-apical to a medio-basal position as they mature. The characterization of hair cells using TEM reveals that hair cells from recently hatched animals contain cuticulosomes that are "diffuse" and around 16 days after hatching they become larger and paracrystalline in structure.Q1) Is it possible that the observed change in size and structure is due to the TEM sample preparation and for some reason younger hair cell cuticulosomes are more susceptible to degradation?

While we cannot exclude this possibility, we think it is very unlikely. For all developmental time points the samples were prepared the same way. Cochlear ducts were dissected from the inner ear, before drop fixing overnight in 2.5% glutaraldehyde supplemented with 2% paraformaldehyde in Soerensen buffer (pH 7.4). Tissues were then incubated in osmium tetroxide for 1 hour, dehydrated in increasing ethanol solutions and embedded in epoxy resin. Moreover, when analysing the samples by TEM there was no evidence of tissue degradation in younger birds.

Next, the authors demonstrate that the cuticulosomes are composed of nanoparticles of ferritin by performing ferritin heavy chain IHC.Q2) Although the ferritin heavy chain-rich puncta resemble the cuticulosomes there is no quantification to demonstrate that these structures develop similar to PB-positive cuticulosomes and hence they are likely the same organelle.

We thank the reviewer for this comment, but we think an additional developmental time series (that would require the preparation of cryosections of n=3 birds, for 5 timepoints) is beyond the scope of this 2-month revision. We are, however, very confident that the ferritin positive structures we report in Figure 4 are actually cuticulosomes. We are able to see cuticulosomes with transmitted light in 10μm thick sections that have been fluorescently stained, particularly in adults where the cuticulosomes are larger. In all cases where we observed punctate ferritin positive structures in the cuticular plate they overlapped with cuticulosomes identified by transmitted light. Moreover, our electron microscopy studies have consistently shown that the cuticulosome is composed of ≈8nm particles that are of the same size and shape of ferritin at all developmental time points (See also Lauwers et al., 2013).

The transcriptomic analysis of hair cells with and without cuticulosomes is done with 1-year-old animals. One of the most highly expressed transcripts is the ferritin heavy chain in cuticulosome positive and negative hair cells.Q3) The ferritin IHC data in Figure 4 seem to indicate that there is diffuse heavy chain expression in all cells in 1-year-old animals. However, this background staining is missing in newly hatched birds. Since this is a developmental study that impinges on the presence of ferritin nanoparticles in cuticulosomes, it is critical to know the profile of the transcriptomic information from newly hatched birds, as well. At the very least, expression of ferritin heavy chain transcript across developmental stages should be confirmed by either in situ hybridization or single cell quantitative RT-PCR.

Our ability to perform single cell qPCR on cuticulosome positive cells is dependent on our ability to pick hair cells with cuticulosomes. As newly hatched birds have few cuticulosomes this experiment is problematic. Moreover, the smaller cuticulosomes at hatching are challenging to identify with transmitted light. As an alternative we have microdissected the distal region of the basilar papillar from multiple developmental stages (hatching, 8 days, 16 days, 30 days, 1 year), extracted mRNA, generated cDNA, and performed qPCR on heavy chain ferritin transcripts. These data are now shown in Figure 4—figure supplement 1. This experiment revealed that the expression of heavy chain ferritin peaks at 16 days of age, but is consistently high at all developmental time points. (subsection “The cuticulosome contains ferritin”).

Differential expression analysis of the transcriptomic data reveals that "genes associated with vesicle trafficking, endocytosis, the formation of membrane bound organelles, and metal binding are differentially expressed in cuticulosome positive hair cells."Q4) It is necessary to confirm at least some of these gene expression profiles with in situ hybridization.

As indicated above (reviewer 1), we have found *in* situ hybridisation in the adult inner ear very challenging. As an alternative we assessed the levels of RAB5B and RNF128 in cuticulosome positive and cuticulosome negative hair cells (n=8 birds) by qPCR. These data which are now shown in Figure 5, confirm that RAB5B is enriched in cuticulosome positive cells and RNF128 is enriched in cuticulosome negative cells. Our attempts to validate additional candidates (such as STXBP5L and CLIV_009848) were unsuccessful as they are expressed at low levels (FPKM<5) and we were not able to amplify them by qPCR (in either cuticulosome positive or negative cells). The text has been updated accordingly, subsection “Transcriptomic analysis of hair cells with cuticulosomes”.

Q5) In their abstract the authors state that "These data suggest that the cuticulosome and the associated molecular machinery regulate the concentration of Fe3+ within the labyrinth of the inner ear, which might indirectly tune a magnetic sensor that relies on electromagnetic induction." However, in their previous paper they have stated that "There is no evidence that sensory neurons play a role in iron homeostasis, and it is not clear why a role in iron storage would necessitate the conspicuous positioning of a single organelle in the cuticular plate." In addition, it is not clear why these structures would need to be composed of ferritin particles that are arranged in a paracrystalline array, heavily regulated developmentally in size/position and appear only in a percent of cells to perform the proposed homeostatic function.

In this manuscript we present data which supports a role of vesicular trafficking in the development and maintenance of the cuticulosome. We agree with the reviewer that it is unclear why hair cells nucleate iron within the cuticular plate in paracrystalline arrays. We cannot (at this stage) make any definitive statements with regards to the function of this organelle, but we think our proposition that it may play a role in the regulation of cation concentrations within the inner ear labyrinth is not unreasonable.

Reviewer #3:The manuscript by Keays and colleagues is the first study that investigates the formation and trafficking of cuticulosomes in pigeon hair cells during early development after hatching. The data revealed in this detailed manuscript are important for understanding the development of magnetoreception in young pigeons. However, a few questions should be addressed before the manuscript is suitable for publication:1) One of the main concerns is the low-number animal cohorts (3-6) and few (e.g. 3) repetitions of each experiment. Was the power analysis ever performed, or what was the rationale for cohort size selection? (There's only one RNASeq experiment for which authors discuss their sample size selection.)

For our histological experiments we did not perform a power analysis. In choosing a n=3-6 for our experiments we were guided by our previous developmental studies on mice (Breuss, Morandell et al., 2015, Breuss, Fritz et al., 2016) and histological work on pigeons (Treiber, Salzer et al., 2012, Lauwers, Pichler et al., 2013, Treiber, Salzer et al., 2013). Moreover, this reflects the standard in the field, where n's of 3-5 are routinely employed to study the development of the avian inner ear (Furutera, Takechi et al., 2017, Jiang, Kindt et al., 2017, Nishitani, Ohta et al., 2017).

2) Subection “Cuticulosomes form rapidly after birds hatch”, percent averages of cuticulosome positive cells are provided with no standard deviation/error. Given the small cohort sizes it would be valuable to see the deviations.

This data is displayed graphically in Figure 1. Dot plots show individual data points and the error bars show the standard error of the mean. In addition we have now inserted the errors in the main text as requested.

3) As authors indicate that the ferritin clusters cuticulosomes may be contributing to magnetoreception, it would be interesting to identify what is the iron content of these structures – that is spacing between individual iron atoms.

We have previously undertaken selected area electron diffraction experiments on cuticulosomes in epon embedded sections. This has yielded diffraction patterns indicative of ferrihydrite ((Fe^3+^)_2_O_3_•0.5H_2_O) (Lauwers, Pichler et al., 2013). Consistent with this finding our Prussian Blue staining has revealed the presence of Fe^3+.^ However, it has been reported that ferritin also contains magnetite (Cowley, Janney et al., 2000, Quintana, Cowley et al., 2004, Gossuin, Hautot et al., 2005). While a specific stain for magnetite (Fe^2+^Fe^3+^_2_O_4_) is not available, Ferric iron (Fe^2+^) (which is a component of magnetite) can be stained with the Turnbull Blue method (Wang, Ong et al. 2002). We have now performed this experiment on our birds at hatching, 8 days, 16 day, 30 days and 1 year. As the staining was very light, indicative of low levels of Fe^2+^, we post stained sections with the chromogen 3,3'-Diaminobenzidine (DAB) which is known to amplify the Turnbull Blue reaction (Wang, Ong et al. 2002). We observed a sharp increase in Turnbull blue staining in the basilar papilla and lagena from hatching to one month of age, mirroring the results of our Prussian Blue staining.

We report that in:

· 1-day-old birds 2.27% of hair cells are TB positive (n=3 birds, n=841 cells) in the basilar papilla and 0.4% of hair cells are TB positive (n=3 birds, n=962 cells) in the lagena macula.

· 8-day-old birds 17.5% of hair cells are TB positive (n=3 birds, n=536 cells) in the basilar papilla and 1.3% of hair cells are TB positive (n=3 birds, n=2007 cells) in the lagena macula.

· 16-day-old birds 29.12% of hair cells are TB positive (n=3 birds, n=796 cells) in the basilar papilla and 1.64% of hair cells are TB positive (n=3 birds, n=2935 cells) in the lagena macula.

· 30-day-old birds are 23.9% of hair cells are TB positive (n=3 birds, n=881 cells) in the basilar papilla and 1.12% of hair cells are TB positive (n=3 birds, n=2978 cells) in the lagena macula.

· 1-year-old birds 19.2%, of hair cells are TB positive (n=3 birds, n=905 cells) in the basilar papilla and 1.16% of hair cells are TB positive (n=3 birds, n=3104 cells) in the lagena macula.

This experiment reveals that the cuticulosome contains Fe^2+^ at all developmental stages, which is consistent with presence of low levels of magnetite. These data are now shown in Figure 4—figure supplement 2, and the text has been modified accordingly (subsection “The cuticulosome contains ferritin”).

4) Following up on the previous point, the magnetic properties of the cuticulosome are of great interest. Have the authors performed vibrating sample or SQUID magnetometry on these structures? Before magnetoreception can be discussed, the structures need to be proven magnetic or conductive (if the inductive mechanism is considered).

We have made a number of attempts to purify cuticulosomes (to enable SQUID analysis), as well as undertaking SQUID analysis on the basilar papilla itself. Despite the pooling of large numbers of samples we have not been able to detect any magnetic signature above background for these experiments – indicating that SQUID lacks the sensitivity we require. As a consequence we are currently in the process of undertaking nanoscale magnetic imaging employing nitrogen valence centers. This method relies on the exchange of magnetisation between nitrogen valence centers on a diamond slide and the magnetic domains of the cuticulosome (Degen, 2008). We have recently encountered some technical issues regarding the preparation of the diamonds. We are therefore not in a position to make any conclusions regarding the magnetic properties of the cuticulosome at this stage. We will aim to publish these findings at a later time.

We wish to emphasize, however, that the inductive mechanism that we propose in the discussion is not dependent on the magnetic properties of the cuticulosome. It is rather dependent on the conductive properties of the endolymph and the presence of a highly sensitive electroreceptor in a population of sensory cells.

5) The authors suggest that the cuticulosome and the associated molecular machinery regulate the concentration of Fe3+ within the labyrinth of the inner ear. However, whether the concentration of Fe3+ changes during development is still unknown. The correlation between Fe3+ concentration in the labyrinth and the expression of cuticulosomes is also not clear. The authors need to provide stronger evidence or supporting references to show the relation between Fe3+ concentration in the labyrinth and the number of cuticulosomes in hair cells.

The reviewer has suggested an experiment that we are currently attempting to undertake. We wish to sample the endolymph at various developmental stages, followed by analysis of its ionic composition. As far as we are aware this experiment has not previously been performed – it is technically very very challenging. It requires that we pierce the bony labyrinth of the inner ear with a glass pipette or needle, sample microliters of the endolymph (while avoiding contamination from the perilymph), followed by analysis by inductively coupled plasma-mass spectrometry. We hope that we will be able to master this protocol, in birds of various ages, in the next 12-24 months.

6) It is surprising that the number of cuticulosomes is the highest in 30-day old pigeons. The authors, however, do not show the localization or the size distribution of cuticulosomes in 30-day old pigeons as compared to 1-year old pigeons. The authors should demonstrate whether the cuticulosomes are fully developed in 30-day old pigeons.

We have undertaken a subcellular analysis of cuticulosomes at 30 days of age, quantitating their size, association with vesicles, presence of membranes and paracrystalline organisation (n=3 birds, n=17 cuticulosomes). These data are now shown in an updated version of Figure 2 and Figure 2—source data 1. It shows that the cuticulosomes at 30 days seem to be on the cusp of maturation. They are of similar size to one-year-old birds (514.3 ± 31 nm at 30days vs 545 ± 31nm in adults), have a similar degree of paracrystalline organisation (17% at 30 days v 16% in adults), but a higher percentage are incomplete (17% at 30 days v 7% in adults).

7) In Figure 4, panels J–L would be better used by showing the immunostaining for Ferritin/Actin at 1 day, 8 days and 1 year for consistency with panels A–I. Showing 3 separate stains at 1 year is redundant (obviously multiple staining experiments must be performed for quantification of expression).

We have rectified this as requested.

8) According to the results presented in this manuscript, cuticulosomes develop within a few days after hatching, and the number of cuticulosomes in the basilar papilla peaks in 30-day old pigeons. This is at odds with prior work that demonstrated that young pigeons (<12 weeks old) are incapable of perceiving magnetic field in the dark (Wiltschko and Wiltschko, 1981). The authors should discuss this in their manuscript.References: Wiltschko, W. and Wiltschko, R., (1981) Disorientation of inexperienced young pigeons after transportation in total darkness. Nature 291: 433 – 434.

We think it is best to exercise caution when assessing behavioural studies in light of the timeline of cuticulosome formation. It should also be noted that Wiltschko and Wiltschko, 1981 did not directly assess the magnetic sense in 12-week-old pigeons, but rather the orientation of outward homing when pigeons were transported in darkness. They note in this manuscript that "we do not know the exact cause of the disorientated behavior of our experimental birds […] as homing experiments with birds is a highly complex behaviour". Nonetheless, we think the reviewer has raised an interesting point and have therefore modified the Discussion as follows:

"The critical question that remains is: What is the function of the cuticulosome? And might it be associated with the magnetic sense? This is a tempting proposition given that pigeons begin to fly at approximately one month of age, a time when the cuticulosome is on the cusp of maturation (Hazard, 1922, Nam and Lee, 2006). However, this correlation should be viewed cautiously as there is conflicting behavioural data supporting the presence of a magnetic sense in young birds. Wiltschko and colleagues have reported that chickens aged 10 days can be conditioned to a magnetic anomaly (Denzau, Kuriakose et al., 2011) and juvenile European Robins are known to possess a magnetic sense (Muheim, Backman et al., 2002), whereas pigeons aged 12 weeks are disorientated if they are transported in the dark (Wiltschko, 1981). An experiment that assessed whether magnetic stimuli results in c-fos associated neuronal activation in pigeons of different ages would be valuable (Wu and Dickman, 2011)."